**EMBO** *reports*

# Robotic platform for microinjection into single cells in brain tissue

Gabriella Shull[1,2,†], Christiane Haffner[3,†], Wieland B Huttner[3] (iD), Suhasa B Kodandaramaiah[1,4,*] (iD) & Elena Taverna[3,5,**] (iD)

## Abstract

Microinjection into single cells in brain tissue is a powerful technique to study and manipulate neural stem cells. However, such microinjection requires expertise and is a low-throughput process. We developed the "Autoinjector", a robot that utilizes images from a microscope to guide a microinjection needle into tissue to deliver femtoliter volumes of liquids into single cells. The Autoinjector enables microinjection of hundreds of cells within a single organotypic slice, resulting in an overall yield that is an order of magnitude greater than manual microinjection. The Autoinjector successfully targets both apical progenitors (APs) and newborn neurons in the embryonic mouse and human fetal telencephalon. We used the Autoinjector to systematically study gap-junctional communication between neural progenitors in the embryonic mouse telencephalon and found that apical contact is a characteristic feature of the cells that are part of a gap junction-coupled cluster. The throughput and versatility of the Autoinjector will render microinjection an accessible high-performance single-cell manipulation technique and will provide a powerful new platform for performing single-cell analyses in tissue for bioengineering and biophysics applications.

**Keywords** brain development; computer vision; neural stem cells; robotics; single cell manipulation

**Subject Categories** Methods & Resources; Neuroscience

## Introduction

Microinjection into cells, in which a glass micropipette is briefly inserted into the cytoplasm or nucleus of a cell to introduce femtoliters of reagents, notably membrane-impermeant ones, is an important tool to manipulate and track single cells and, if applicable, their progeny [1–4]. Recently, microinjection has been adapted to target single neural stem cells in organotypic slices from the developing brain tissue, where it offers several unique advantages [5,6]. First, thanks to its excellent single-cell resolution, microinjection of fluorescent dyes allows correlating single-cell behavior as observed upon live imaging with tissue morphogenesis. Second, microinjection provides flexibility with regard to the chemical composition and complexity of the solution delivered into the cells. For example, pools of RNAs, or even an entire transcriptome, can be delivered into a cell, which allows for the combinatorial analysis of genes affecting brain development [6–9]. Moreover, unlike electroporation, microinjection enables the delivery of both charged and non-charged molecules. Finally, recent work has shown that microinjection can target neural stem cells from multiple species [5] and can be used for CRISPR/Cas9-mediated disruption of gene expression [7].

Despite the advantages it offers, microinjection suffers of several limitations: It is a low-throughput and low-yield process, and it requires a high level of skill and significant practice to master. Ideally, the stereotyped operation of precisely steering the microinjection pipette to cells while visualizing the pipette and tissue under microscope guidance can be implemented by a robotic system, so that sources of variability such as the depth of microinjection, the spacing between injections, and the volume of solution delivered to the cells can be precisely controlled. Such a robot would greatly increase throughput and yield of microinjection in tissue, opening this technology to a greater user base within neuroscience, developmental biology, cell biology, and biophysics.

Here, we report the development of an image-guided microinjection robot, the "Autoinjector", that utilizes images acquired from a microscope to guide the microinjection needle into single cells in tissue at controlled pressure with micrometer-scale precision. This process can be repeated to target hundreds of cells, thereby increasing the rate and success of microinjection by an order of magnitude as compared with manual operation. The Autoinjector allowed us to target neural stem cells and follow their lineage progression in culture over time. The Autoinjector was also used to study the

1  Department of Biomedical Engineering, University of Minnesota, Twin Cities, MN, USA
2  Department of Biomedical Engineering, Duke University, Durham, NC, USA
3  Max Planck Institute of Molecular Cell Biology and Genetics, Dresden, Germany
4  Department of Mechanical Engineering, University of Minnesota, Twin Cities, MN, USA
5  Max Planck Institute for Evolutionary Anthropology, Leipzig, Germany
   *Corresponding author. Tel: +1 612 301 1636; E-mail: suhasabk@umn.edu
   **Corresponding author. Tel: +49 341 3550; E-mail: elena_taverna@eva.mpg.de
   †These authors contributed equally to this work

cell-to-cell communication in the developing mouse telencephalon. We focused our attention on gap-junctional coupling in neural stem and progenitor cells and found that coupled clusters contain both apical and basal progenitors, the two main classes of stem and progenitor cells in the mouse developing brain. Finally, we made use of the micrometer-scale precision and the flexibility of the robot by targeting single newborn neurons in organotypic slices from the mouse and human developing brain, a result never achieved before. The Autoinjector can be implemented on any standard microscope setup and will be a valuable resource for developmental neurobiologists to study brain development. Not limited to the developmental biology field, the Autoinjector will enable the quantitative analysis of single-cell behavior in brain tissue in developmental biology, cell biology, and biophysics.

## Results

### The Autoinjector—a robot for image-guided microinjection into single cells in brain tissue

A typical microinjection experiment involves a user guiding the injection micropipette to the surface of the tissue, inserting the pipette tip into the tissue while the micropipette is held under positive pressure to perform the injection, and withdrawing the micropipette back within a brief period. The depth of tissue penetration and time the micropipette stays inserted inside the cell affect the efficacy of microinjection. Both these parameters are highly dependent on individual experimenter's skill and experience and are thus prone to inconsistencies, thereby leading to low yield [5,6]. Lastly, the procedure is extremely tedious to perform. All these hurdles prevent users from injecting a large number of cells, making microinjection into cells in tissue challenging to be used as a robust tool to track and manipulate cells.

We built a robot, the "Autoinjector", which can precisely control the position of the injection micropipette using microscope image guidance (Fig 1). The Autoinjector requires relatively simple modifications to a conventional microinjection station. The injection micropipette is attached to a micropipette holder with a pressure inlet which is mounted on a manipulator for programmatic three-axis position control (Fig 1A). The manipulator is integrated into an inverted microscope (Fig 1A and C), and the pressure inlet is connected to a custom pressure regulator for precise pressure control (Fig 1B and C). Images acquired from the microscope camera are used by an algorithm to guide the injection micropipette to precise locations in the microscope field of view (FOV; Fig 1C, and 2A and B; see also User Manual).

A Movie illustrating the operation of the Autoinjector using organotypic slice cultures of embryonic day 16.5 (E16.5) mouse telencephalon is shown in Movie EV1. First, the magnification of the microscope objective is set by the user in the graphic user interface (GUI, Appendix Fig S1). The injection micropipette is brought into the microscope FOV and imaged along with the tissue (Fig 2A and B top, 2C.i). This is followed by a calibration step, where displacement of the injection micropipette in three dimensions is projected onto the corresponding displacement in the two-dimensional microscope image (Appendix Fig S2A, Appendix Note S1). Once the calibration step is completed, the Autoinjector can guide the injection micropipette to specific locations in the FOV

using the micromanipulator, similar to previous algorithms [10,11]. The user then draws a line along the desired path of microinjection on the microscope image using the graphical user interface (GUI; Fig 2B bottom, 2C.iii; see also User Manual). This is followed by specifying the starting point of the injection micropipette by clicking the tip of the microinjection pipette in the GUI (Fig 2B, bottom, 2C.iv). The algorithm then computes a trajectory (Appendix Note S2) based on the depth into tissue the injection micropipette penetrates for each microinjection attempt, the distance the micropipette is pulled out of the tissue after an injection attempt, and the spacing between subsequent microinjection attempts. Each of these parameters can be independently specified by the user in the GUI (Appendix Figs S1 and S2, Appendix Note S2). Next, the Autoinjector positions the tip of the injection micropipette at the surface of the tissue, inserts the injection micropipette into the tissue to perform a microinjection, retracts the injection micropipette, and positions the injection micropipette at the next location along the path (Fig 2C.v, Movie EV1). This process is repeated until microinjections are completed along the entire path annotated by the user. A constant user-defined pressure is applied to the injection micropipette using the pressure controller throughout this process (see also User Manual).

Prior to microinjection experiments, we investigated the Autoinjector's ability to target locations specified by the user in the microscope FOV. The experimenter annotates the tip of the microinjection pipette at various steps during calibration. Differences in individual perception of the micropipette tip may lead to systematic errors in positioning the pipette after calibration. To test for these effects, three experimenters with no prior experience using the Autoinjector performed calibration (Appendix Note S3). The angle between the camera FOV reference axes and the micropipette reference axes, a key parameter used in performing the transformation between the two coordinate systems (Appendix Note S1), was not significantly different when calibration was performed by either experimenter ($P = 0.400$, $P = 0.700$, $P = 0.800$ for experienced user vs. inexperienced user 1, 2, and 3, respectively). To quantify the spatial error of the Autoinjector, pipettes were calibrated by the expert experimenter and commanded to eight locations spread across 75% of the FOV which represents the area where the targeted tissue is located. The error in image-guided positioning of the pipette was $0.211 \pm 0.182$ μm along the x axis and $0.345 \pm 0.415$ μm along the y axis of the manipulator ($n = 8$ locations, five measurements per location). This error in spatial positioning is much smaller than dimensions of the trajectories (10's to 100's of μm) used and was thus sufficient for image-guided targeting of single cells across the samples.

### Optimizing automated microinjection

Having established that the injection micropipette could be guided to locations in the microscope FOV with micrometer-scale precision, we attempted injection of fluorescent dye into apical progenitors (APs) in organotypic slices of the E14.5 mouse telencephalon and optimized the parameters for automated microinjection (Fig 3 and Movie EV2). APs are important cells for the generation of mature neurons in the neocortex and are key to understanding how higher level cognitive functions evolved in mammals [12–15]. Injecting APs with dye and tracking their progeny provides a good model for studying stem cell biology and cell fate specification in tissue [5–7]. The apical plasma membrane of APs faces the ventricle (Fig 2A),

                                                      

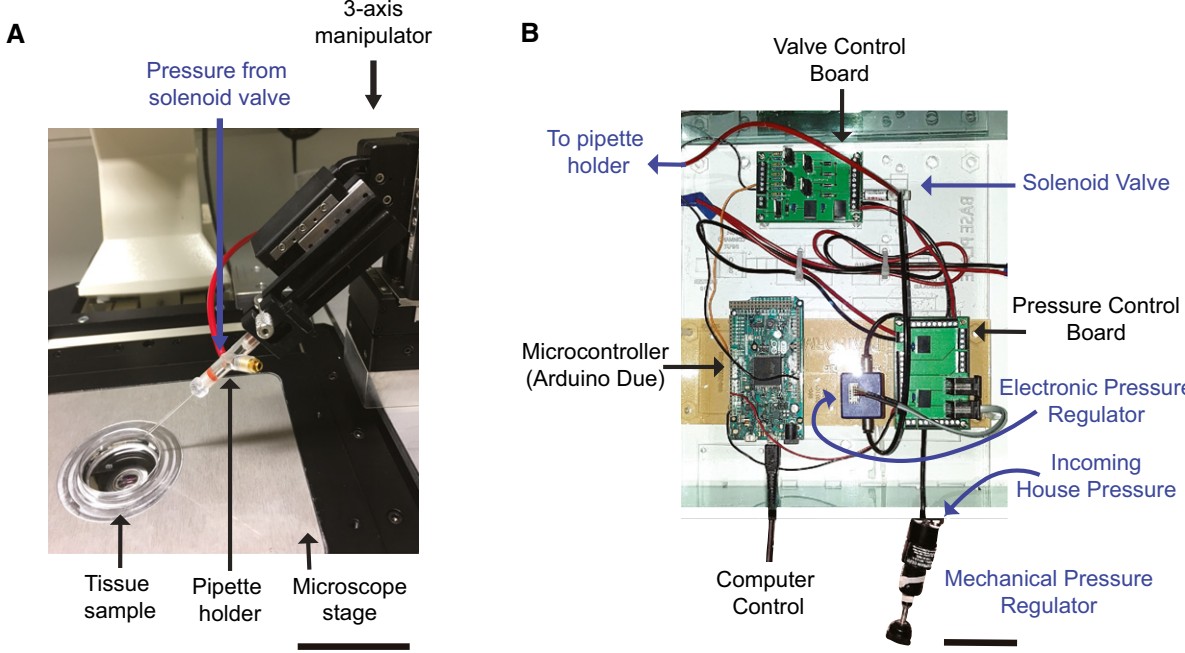

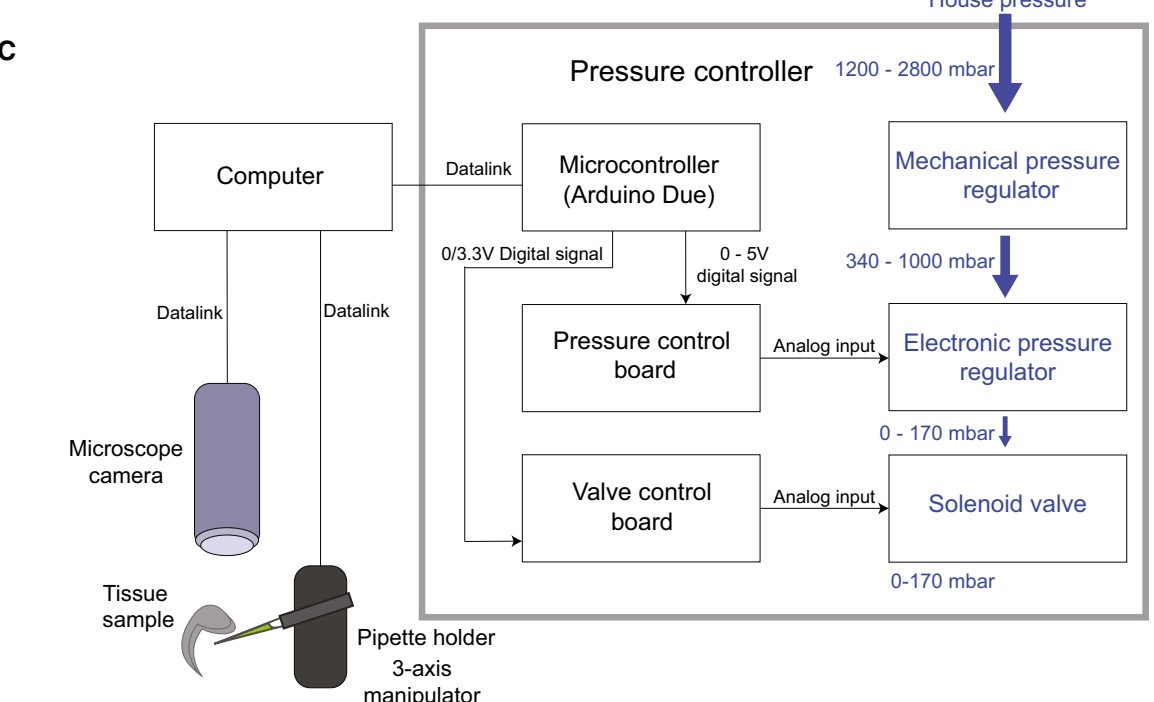

**Figure 1.  Autoinjector—an image-guided microinjection platform.**

A  Photograph of the manipulator end of the Autoinjector. Scale bar is 40 mm.
B  Photograph showing the custom pressure controller used for programmatic pressure control during automated microinjection. Scale bar is 8 mm.
C  Overall hardware schematic. A computer interfaces with all the components of the platform, including the pressure controller, manipulator, and microscope camera. Images from the microscope camera are used to control the position of the micropipette using the manipulator. The pressure controller is used to precisely deliver injection pressure to the micropipette during microinjection. House pressure is coarsely downregulated by a mechanical regulator, followed by fine downregulation using an electronic regulator controlled by a microcontroller. Delivery of pressure to the micropipette is digitally controlled using a solenoid valve. The trajectory of the micropipette is controlled by the manipulator and is guided using images acquired using the microscope. Black arrows indicate digital interface routes, and blue arrows indicate pneumatic route.

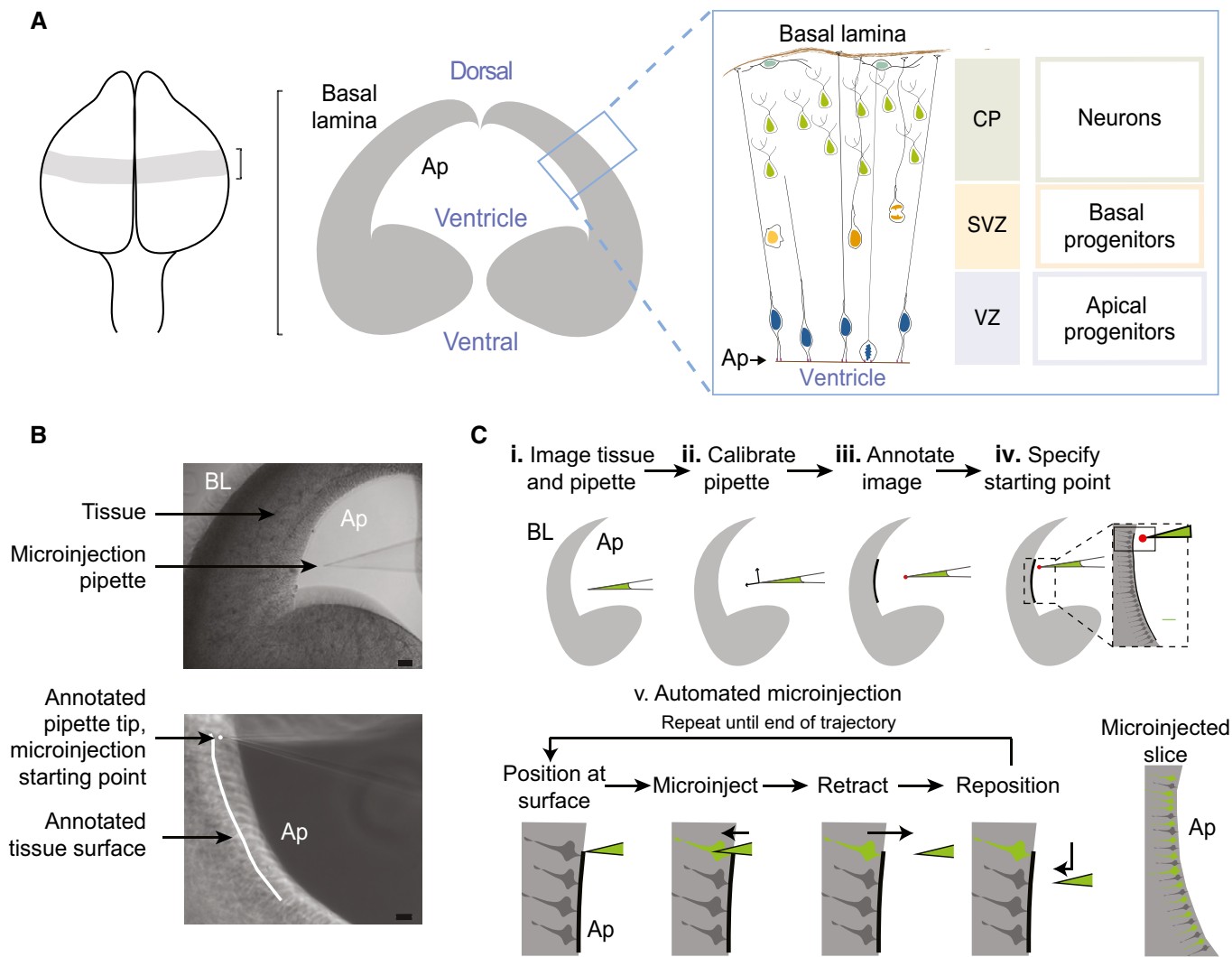

**Figure 2.  Image-guided microinjection protocol and model system.**

A   Schematic of targeted telencephalon: Left, E14.5 brain; middle, coronal section of the brain; right, inset illustrating the organization and major cell types of the cortical wall. APs attached to the ventricle are targeted by automated microinjection.

B   Microscope images of the tissue and microinjection pipette before (top) and after (bottom) annotation of the tissue surface and microinjection micropipette tip by the experimenter. Scale bars: top 100 μm, bottom 10 μm. Ap indicates the apical surface, and BL indicates the basal lamina.

C   A cartoon schematic of the microinjection protocol. (i) The user brings the micropipette into the microscope FOV close to the tissue. (ii) The manipulator is calibrated to allow for image-guided positioning of the micropipette using a calibration algorithm (see Appendix Note S1, and Appendix Fig S2 for additional details). (iii) The user annotates image with the desired trajectory of microinjection by dragging a cursor over the edge of the tissue and clicks a point indicating the tip of the micropipette. (iv) Before injection begins, the user brings the tip of the micropipette close to the tissue surface and specifies the starting point. (v) The following steps are fully automated. The Autoinjector positions the micropipette at the surface of the tissue and advances the micropipette to a specified depth under pressure into the tissue resulting in microinjection. The micropipette is then retracted out of the tissue and repositioned at the next microinjection site. This is repeated until the micropipette reaches the end of the microinjection trajectory. The user can independently specify the depth of microinjection, retraction, and spacing between microinjections (see Appendix Note S2, and Appendix Fig S2 for additional information). Ap indicates the apical surface, and BL indicates the basal lamina.

and their nuclei are found at distances ranging from 5 to 150 μm from the apical plasma membrane depending on the phase of cell cycle [14–16]. During microinjection, we approached APs from the ventricular surface. An ideal microinjection attempt targets the apical process of the APs injecting roughly 10% of the cytosolic volume [6]. Experimenters modulate the internal pressure of the injection micropipette between 75 and 125 mbar pressure during microinjection [5,6]. This internal pressure is a key determinant of the injection volume. We used this range as a heuristic starting

point and investigated how the internal pressure of the injection micropipette affected microinjection yield (Fig 3A and B). Microinjections were performed at a depth of 10 μm from the ventricular surface. Microinjections performed at 75 mbar resulted in 35% of cells successfully injected ($n = 314$ attempts total, two slices). When performed at 100 mbar, 35% of cells were successfully injected ($n = 185$ attempts total, two slices), whereas when microinjections performed at 125 mbar resulted in 8% of cells successfully injected ($n = 287$ attempts total, two slices). Given that there was no

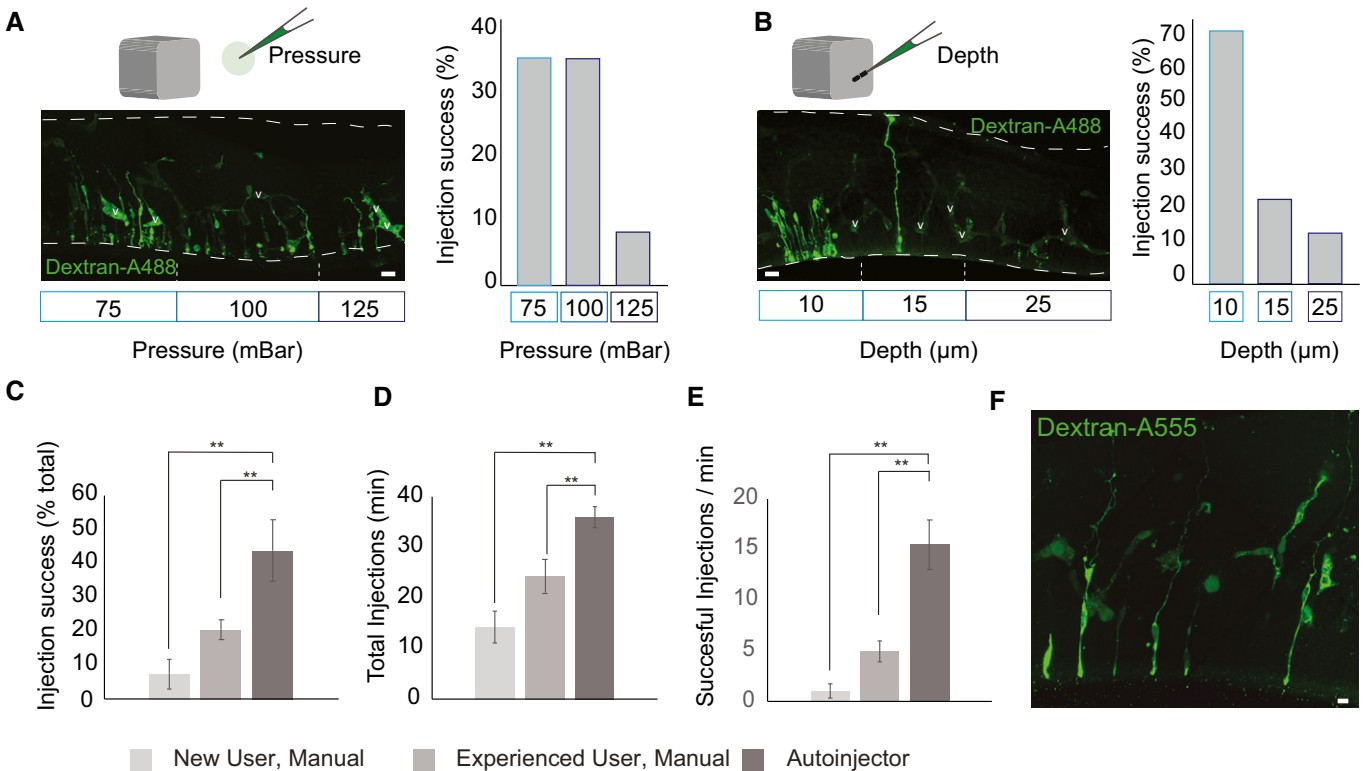

**Figure 3. Autoinjector performance.**

A  (Left) Confocal image of a section of mouse telencephalon fixed and stained immediately after microinjection. Dashed lines indicate locations in the slice where microinjections were attempted at different pressures. Microinjections were attempted at pressure of 75 mbar (left area), 100 mbar (center area), and 125 mbar (right area). (Right) Percentage of successful injections achieved by the Autoinjector at 75 mbar (35%, $n$ = 314 total, 110 successful), 100 mbar (35%, $n$ = 185 total, 65 successful), and 125 mbar (8%, $n$ = 287 total, 23 successful).

B  (Left) Confocal images of a section of mouse telencephalon fixed and stained immediately after microinjection. Dashed lines indicate locations in the slice where microinjections were attempted at different depths keeping the pressure constant at 75 mbar. Microinjections were attempted at depths of 10 μm (left area), 15 μm (center area), and 25 μm (right area) into the apical surface. (Right) Percentage of successful injections achieved by the Autoinjector at depths of 10 μm (68%, $n$ = 163 total, 111 successful), 15 μm (22%, $n$ = 170 total, 37 successful), and 25 μm (11%, $n$ = 169 total, 19 successful) into the apical surface.

C  Successful injections represented as a percentage of total for a novice user on the manual microinjection system, an experienced user on the manual microinjection system, and the Autoinjector.

D  Total injection attempts per minute for a novice user on the manual microinjection system, an experienced user on the manual microinjection system, and the Autoinjector.

E  Successful injections per minute for a novice user on the manual microinjection system, an experienced user on the manual microinjection system, and the Autoinjector.

F  Cells targeted at a constant spacing of 30 μm, scale bar is 10 μm.

Data information: In (A, B), scale bars are 20 μm and v indicates a vessel. For (C–E), $n$ = 190 attempts, five slices for novice user, $n$ = 181 attempts, four slices for an experienced user, and $n$ = 864 attempts, four slices for a novice user on the Autoinjector. Significance was tested using a Mann–Whitney $U$-test, error bars indicate standard deviation, and **$P$ < 0.001.

Source data are available online for this figure.

significant difference between the microinjection efficiency obtained when applying an internal pressure set at 75 mbar vs. 100 mbar, we decided to use a pressure of 75 mbar pressure to minimize the stress applied to the cells and to the tissue during microinjection.

We next assessed how the depth to which the pipette penetrates tissue affects microinjection yield. The Autoinjector was programmed to axially penetrate to depths of 10, 15, and 25 μm from the assigned ventricular surface for microinjection (Fig 3B) with the internal pressure of the micropipette set to 75 mbar. With a 45° approach angle relative to the normal to the tissue surface, this translated to 5, 11, and 18 μm depths from the assigned ventricular surface. This range of depths accounted for variation in cell locations and potential deformations of the ventricular surface during microinjection. Microinjections performed at depths of 10 μm resulted in 68% of cells successfully injected ($n$ = 163 microinjection attempts, two slices). Microinjections performed at depths of 15 μm resulted in 22% of cells successfully injected ($n$ = 170 microinjection attempts, two slices). Microinjections performed at depths of 25 μm resulted in 11% of cells successfully injected ($n$ = 169 microinjection attempts, two slices). For the subsequent experiments, we used an injection depth of 10 μm for injecting APs unless otherwise stated. Robotic control of pressure and position thus allowed us to systematically explore parameters affecting microinjection yield.

## Comparison of manual and automated microinjection

We next compared the performance of the Autoinjector to manual microinjections performed by a novice experimenter (no prior microinjection experience) and an experienced experimenter (5 years of microinjection experience, Fig 3C–E). The Autoinjector was operated by a user with no prior microinjection experience. The optimized depth and pressure parameters derived above were used in these experiments (pressure = 75 mbar, depth = 10 μm). A successful microinjection was indicated by co-localization of the microinjected dye, Dx3-Alexa488, and DAPI, in slices fixed immediately after the experiment (Appendix Fig S3 and Movie EV3). We found that a novice experimenter performing manual microinjection had a success of $7.46 \pm 4.26\%$ with an injection rate of $14.15 \pm 3.13$ attempts/min ($n = 190$ attempts, five slices). This corresponds to a successful microinjection rate of $1.09 \pm 0.67$ injections/min. An experienced experimenter performing manual microinjection had a success of $20.41 \pm 2.91\%$ with an injection rate of $24.14 \pm 3.38$ attempts/min ($n = 181$ attempts, four slices). This corresponds to a successful microinjection rate of $4.95 \pm 1.05$ injections/min. A novice user performing microinjection using the Autoinjector had a success of $43.73 \pm 9.11\%$ (Figs 3C and EV1) with an injection rate of $35.92 \pm 2.12$ attempts/min ($n = 864$ attempts, four slices; Fig 3D). This corresponds to a successful microinjection rate of $15.52 \pm 2.48$ injections/min (Fig 3E). We found that the experienced user had a significantly higher injection success, and rate compared to a novice user using the manual microinjection (see Materials and Methods, $P = 0.004$), the novice user using the Autoinjector had a significantly higher injection success, and rate compared with the novice user on the manual system (see Materials and Methods, $P = 0.0079$), and the novice user using the Autoinjector had a significantly higher injection success, and rate compared with the experienced user on the manual system (see Materials and Methods, $P = 0.0079$). The novice experimenter using the Autoinjector achieved a 15-fold increase in successful injection rate relative to a novice experimenter performing manual microinjections, and a 3-fold increase in successful injection rate relative to an experienced experimenter performing manual microinjections. The increase in injection rate enabled by the Autoinjector represents a significant improvement in yield (Fig 3F; see also Fig EV1).

We next assessed if robotic microinjection affected cell viability by quantifying cell death (Appendix Fig S4) and the progression through cell cycle (Appendix Fig S5). Cell death was assessed by counting the number of picnotic nuclei in the VZ and SVZ and by performing cumulative EdU labeling. We first assessed if the cell viability was affected by the slice culture procedure by comparing the tissue *in vivo* ($n = 5$ sections) with slices that did not undergo microinjection (non-injected slices, $n = 5$ slices). We found that the slice culture procedure increases the number of picnotic nuclei as compared to tissue *in vivo* ($P = 0.004$ using Wilcoxon rank-sum test see Materials and Methods and Appendix Fig S4). This is to be expected because the tissue slicing can be a traumatic process compromising cell viability. We next assessed if the cell viability was affected by the microinjection process. We compared the number of picnotic nuclei in non-injected slices ($n = 5$ slices) and in slices that underwent manual ($n = 7$ slices) or automated

microinjection ($n = 6$ slices). We did not observe any significant difference in the number of picnotic nuclei when comparing non-injected with manually injected slices ($P = 0.6806$, using Wilcoxon rank-sum test see Materials and Methods and Appendix Fig S4), nor when comparing non-injected with slices injected with the Autoinjector ($P = 0.6688$, using Wilcoxon rank-sum test see Materials and Methods and Appendix Fig S4).

Cell viability and progression through cell cycle were also assessed using 24 h EdU cumulative labeling (Appendix Fig S5). We saw no difference between the EdU incorporation in non-injected slices ($n = 4$ slices) and *non-injected cells* in a manually microinjected slice ($n = 1$ slice, $P = 0.6000$, using Wilcoxon rank-sum test see Materials and Methods and Appendix Fig S5) and *non-injected cells* in an automated microinjected slice using the dye alone ($n = 3$ slices, $P = 0.5714$, using Wilcoxon rank-sum test see Materials and Methods and Appendix Fig S5). We saw no difference between the EdU incorporation in non-injected slices ($n = 4$ slices) and *injected cells* in a manually microinjected slice ($n = 4$ slices, $P = 0.5571$, using Wilcoxon rank-sum test see Materials and Methods and Appendix Fig S5) and *injected cells* in an automated microinjected slice using the dye alone ($n = 3$ slices, $P = 0.5714$, using Wilcoxon rank-sum test see Materials and Methods and Appendix Fig S5), or the dye and RFP mRNA ($n = 1$ slice, $P = 0.4000$, using Wilcoxon rank-sum test see Materials and Methods and Appendix Fig S5). Based on these observations, we can conclude that the Autoinjector does not compromise tissue and cell viability.

## Autoinjector allows tracing cell fate transition and lineage progression of neural stem and progenitor cells in tissue

Microinjection is a useful tool to track neural stem and progenitor cells, and their progeny in organotypic slice culture as individual cells can be labeled in a sparse and spatially defined fashion [6]. We used the Autoinjector to inject APs along the ventricular surface and tracked the location, morphology, and cell identity of the injected cells and their progeny after 0, 24, and 48 h in culture (Figs 4 and EV2). The injected cells and their progeny were scored based on the distance of the cell body from the ventricular surface and based on the layer in which they reside [ventricular zone (VZ), subventricular zone (SVZ), or cortical plate (CP), Fig 2A]. Cell identity was determined by assessing the positivity for Sox2, a transcription factor typically expressed by APs (Fig EV2), T-box brain protein 2 (Tbr2), a transcription factor characteristically expressed by basal intermediate progenitors (bIPs) in mouse [17], and for the neuron-specific class III β-tubulin (TuJ1; Fig 4A–C) [18]. At time zero, 100% of the injected cells ($n = 147$ cells total) were in the VZ (Fig 4D) and were bearing an apical attachment (not shown). After 24 h in culture, 31% of the progeny of injected cells ($n = 151$ cells total) were in the VZ, while 59% were found in the SVZ, and 10% in the CP. After 48 h in culture, 23% of the injected cells ($n = 26$ cells total) were in the VZ, 31% in the SVZ, and 47% in the CP. The redistribution of the injected cells and their progeny to a more basal compartment was paralleled by an increase in their distance from the apical surface (Fig 4E) and by a change in cell identity (Fig 4F and G). At 0 h, the cells targeted by microinjection were all TuJ1 negative ($n = 13$ cells total). The majority of the microinjected cells was Sox2-positive (88%; $n = 83$ cells

total; Figs 4A–C and EV2). A fraction of the microinjected cells was Tbr2 positive (10.38%; $n = 77$ cells total; Figs 4A–C and EV2), consistent with previous results [5] and with the notion that a newborn basal progenitor (BP) retains an apical contact when it is generated [19] and can be therefore targeted by microinjection. After 24 h in culture, 62% of the cells were

Sox2-positive, ($n = 53$ cells in total), 32% of cells were Tbr2-positive, and 33% were TuJ1-positive ($n = 92$ cells total), indicating a shift toward a more basal cell identity (Figs 4A–C and EV2). After 48 h in culture, 38% of the cells were Sox2-positive, ($n = 21$ cells in total), 50% of cells were Tbr2-positive, and 66% were TuJ1-positive ($n = 18$ cells total), indicating the continuous progression of

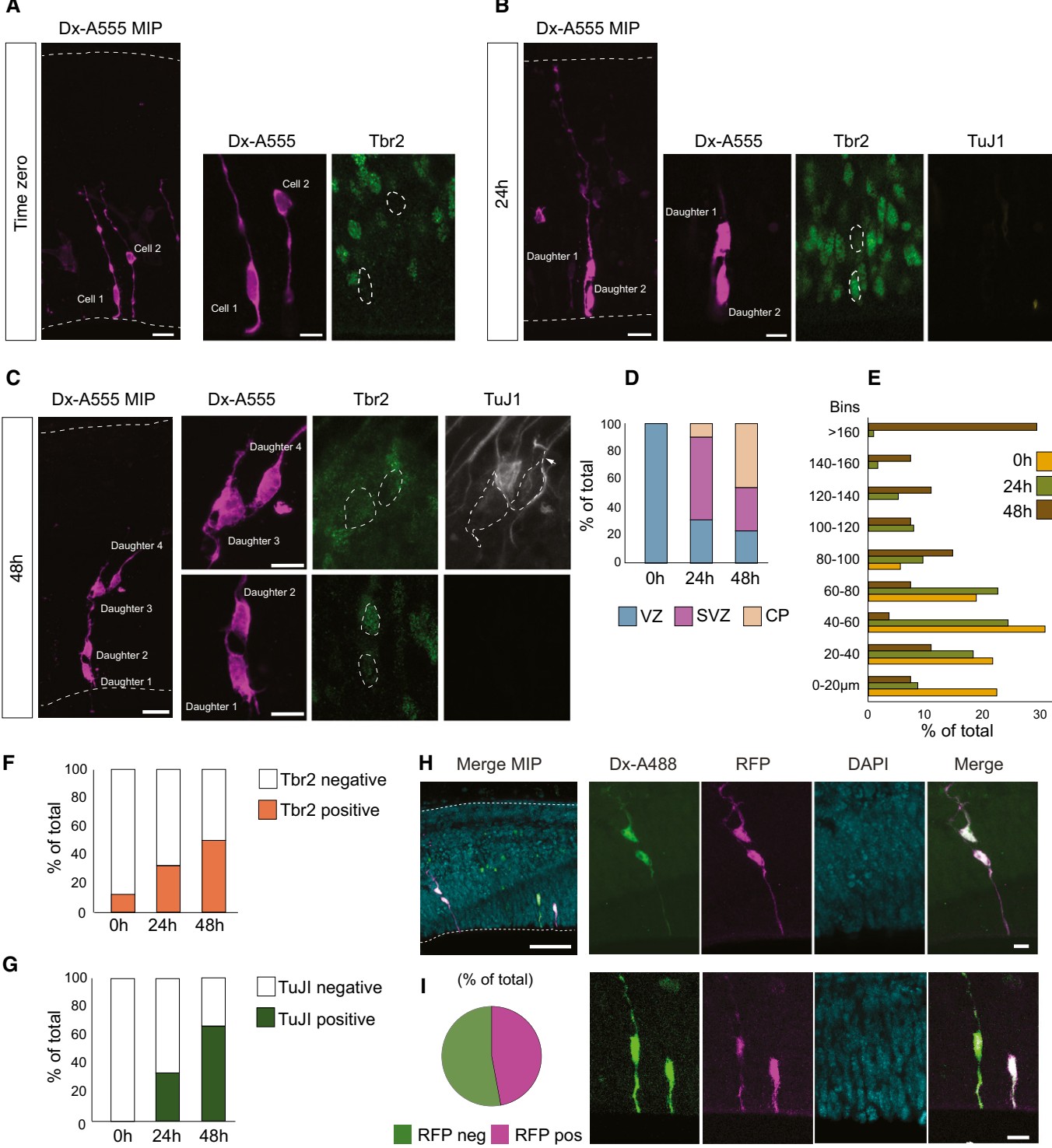

**Figure 4.**

**Figure 4.  Neural stem cell lineage tracing and expression of exogenous mRNA.**

Automated microinjection was performed on organotypic slices of mouse E14.5 dorsal telencephalon using Dextran-A555 (A–G) or Dx-A488 along with mRNA for RFP (H–I), without (0 h, A, D–G) or with slice culture for 24 or 48 h (24 h, B, D–G, H, I; 48 h, C, D–G). After fixation, slices were stained for Tbr2 and TuJI. (A–C) Fluorescence images of tissue after microinjection with Dextran-A555 (magenta). The tissue was stained for Tbr2 (green) and TuJI (white).

A    Microinjected cell at 0 h; injected cells show the typical bipolar morphology of an AP and are mainly negative for the BP marker Tbr2.
B    Two-daughter cell progeny 24 h after microinjection; the top cell shows the bipolar morphology characteristic of an AP and is negative for Tbr2; the bottom cell is a newborn BP positive for Tbr2. Both cells are negative for the neuronal marker TuJI (white).
C    Four-daughter cell progeny 48 h after microinjection; note that daughter 1 and 2 reside in the VZ, while daughter 3 and 4 are positive for Tbr2 and TuJI and reside in the SVZ.
D    Distribution of microinjected cells and their progeny in the VZ, SVZ, and CP (cortical plate). n = 147 cells total for 0 h, 151 cells total for 24 h, and 26 cells total for 48 h.
E    Distribution of microinjected cells into nine bins based on the distance from the ventricular surface (0 µm) and expressed as % of total. n = 147 cells total for 0 h, 151 cells total for 24 h, and 26 cells total for 48 h.
F    Expression of Tbr2 in microinjected cells and their progeny at 0, 24, and 48 h. n = 147 cells total for 0 h, 151 cells total for 24 h, and 26 cells total for 48 h.
G    Expression of TuJI in microinjected cells and their progeny at 0, 24, and 48 h. n = 147 cells total for 0 h, 151 cells total for 24 h, and 26 cells total for 48 h.
H    Organotypic slices of mouse E14.5 dorsal telencephalon injected with Dextran-A488 (green) and RFP poly-A+-mRNA. After 24 h in culture, the progeny expresses RFP (magenta; blue; DAPI). n = 17 cells injected, 8 RFP positive.
I    Percentage of RFP-positive and RFP-negative cells (n = 17 cells injected).

Data information: In (A–C, H), the images on the left are maximum intensity projections (MIP) of 27, 50, 47, and 60 focal planes, respectively; images on the right are single optical sections corresponding to the nucleus area; scale bars are 20 µm for MIP and 10 µm for single focal planes. In (H), the scale bar in the top left image is 100 µm. VZ, ventricular zone; SVZ, subventricular zone; CP, cortical plate.
Source data are available online for this figure.

cells toward BP and neuronal cell identity (Figs 4A–C and EV2). Thus, the assessment of localization, distance, and cell identity at 0, 24, and 48 h after microinjection suggests that the Autoinjector does not interfere with the normal ability of the APs to generate downstream progeny, represented by BPs and neurons. Taken together, the data shown here demonstrate that Autoinjector allows tracing cell fate transition and lineage progression of neural stem and progenitor cells in tissue.

### Delivery of exogenous mRNA via microinjection

Genetic manipulation of neural progenitors is an invaluable tool to investigate the genetic basis of neocortex development and evolution [9,15,20,21]. Injection of mRNA of genes of interest into cells allows the effects of these genes on neocortex development to be quantified [5,6,8,9]. We used the Autoinjector to inject the mRNA for red fluorescent protein (RFP) into cells along with an injection dye and observed the ability of the injected cells and their progeny to express the RFP after 24 h in organotypic slice culture. Of the cells that were injected, 47% expressed RFP after 24 h in culture (n = 17 cells in total) as indicated by the co-localization of the injection dye and the presence of RFP fluorescence (Fig 4H and I). The yield of RFP translation with the Autoinjector represents an improvement compared with previously reported translation yield (≈ 20%) using manual microinjection [6].

### Autoinjector allows a quantitative dissection of cell-to-cell communication in the developing brain

We used the Autoinjector to gain insight into cell-to-cell communication in the developing brain, focusing on gap junction communication. Gap junctions are intercellular channels that allow direct diffusion of ions and small molecules between adjoining cells and are thought to play important roles in development including neuronal differentiation, migration, and circuit organization [22–25]. We used the Autoinjector to microinject a combination of gap junction permeable fluorescent dye (Alexa-488) and gap junction-impermeable fluorescent dye (Dx-3000-Alexa-555) into APs. To

identify unique clusters of coupled cells, we programmed the Autoinjector to perform injections on cells separated by 30 µm. The slices were fixed immediately after microinjection to make sure we were detecting clusters of coupled cells and not clusters of daughter cells forming as a consequence of lineage progression. This experimental paradigm allowed us to quantitatively determine (i) how many neural stem cells are coupled via gap junctions, (ii) the size and distribution of coupled clusters, and (iii) the contribution of different neural stem cell types to the coupled clusters (Fig 5).

We found that at mid-neurogenesis in the mouse developing brain one-third of the targeted cells are part of a coupled cluster (Fig 5A–D). The cluster size is variable and ranged from 2 to 8 cells (Fig 5E). The majority of clusters are two-cell clusters (Fig 5E). Of note, in all two-cell clusters the junction-impermeable dye (Dx-3000-Alexa-555) labels only one cell (Appendix Fig S6), ruling out the possibility of a daughter cell pair connected by a midbody bridge and confirming that the two cells are coupled via gap junctions. Furthermore, coupled cells are confined to the VZ (Fig 5F and G) and tend to occupy a more apical position compared with non-coupled cells. This observation is consistent with the idea that coupling is cell cycle dependent and that, at least in late neurogenesis, cells in S-phase are less likely to be part of a coupled cluster [25].

To assess the identity of the coupled cells, we next stained the tissue for Tbr2, a marker for mouse BPs (Fig 5H and I). We found that only a minor proportion of coupled cells (9%) are Tbr2-positive (n = 46 cells total; Fig 5J). If one considers the entire VZ, then > 30% of the cells in the VZ are Tbr2-positive (Fig 5J) [26,27]. This discrepancy suggests that the coupled clusters contain a sub-population of Tbr2-positive cells. From a cell biological point of view, the VZ contains two sub-populations of Tbr2-positive cells: one that keeps an apical contact (non-delaminated) and one that lost it during delamination (delaminated) [19]. We checked for polarity cues in coupled cells, and we found that the vast majority of cells (98% of cells) in coupled clusters have an apical process and therefore maintain an apical polarity cue (Fig 5K). These data suggest that the maintenance of the apical contact is a crucial factor influencing the ability of a Tbr2-positive cell to be part of a cluster. Taken together, these results demonstrate that the Autoinjector can be

used to gain insight into the fine cell biology of neural stem cells in organotypic slices of the developing brain.

## Targeting newborn neurons in the developing mouse and human telencephalon

Finally, we wanted to determine if automated microinjection could be applied in a generalized manner to different cell types in the developing mouse brain, and if it could be performed in tissues from other species. We first tested the ability of the Autoinjector to target APs in the hindbrain, the tissue where manual injection was originally developed [6]. As shown in Appendix Fig S7, the Autoinjector successfully targets hindbrain APs, highlighting their morphology. We then used the Autoinjector to microinject newborn neurons in organotypic slices. Neurons are extremely challenging to target with manual microinjection because of their distance from the basal surface and the topology of the tissue (Fig 6A). In order to target newborn neurons, the Autoinjector approached the tissue slice from

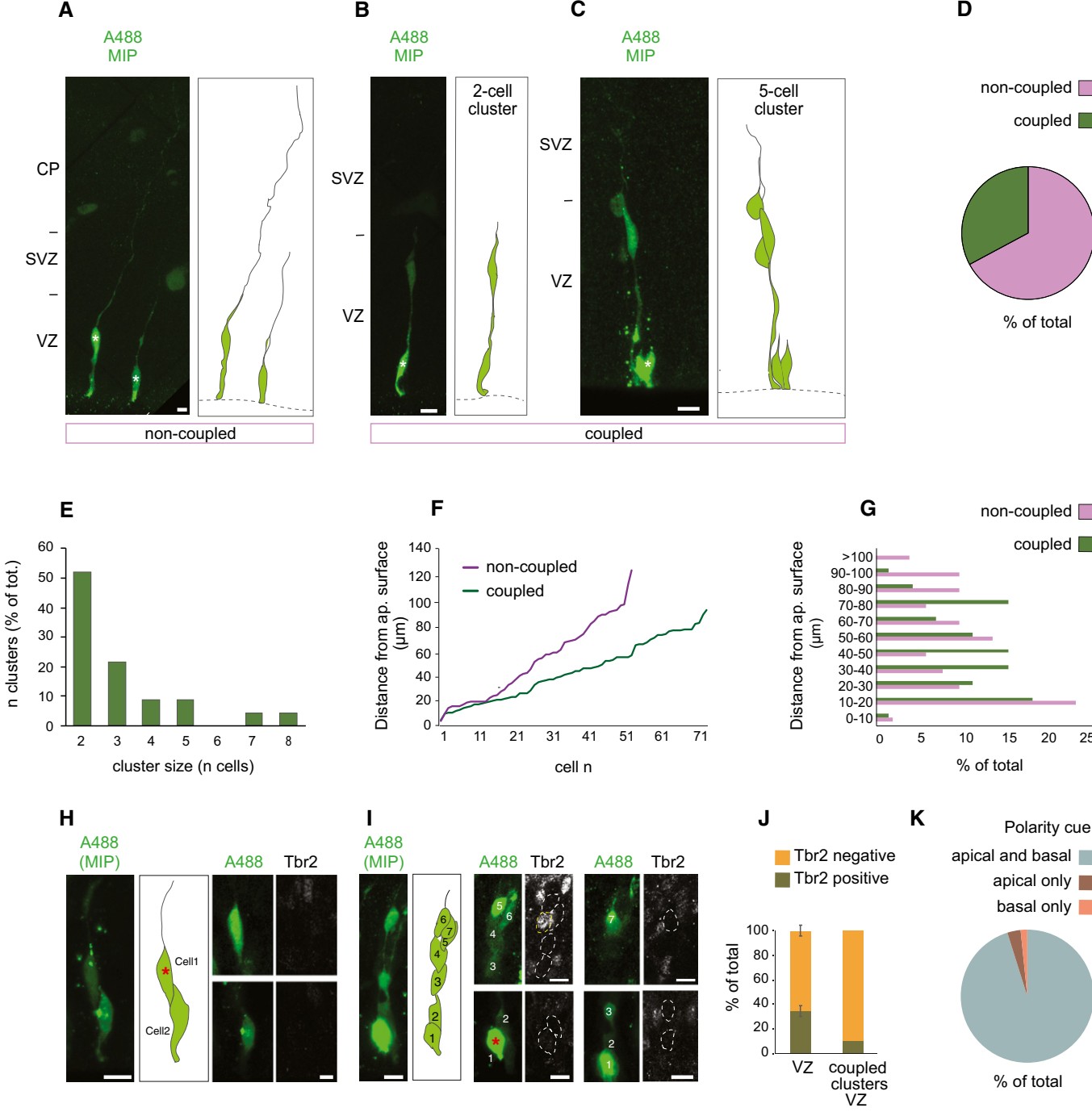

**Figure 5.**

**Figure 5. Quantitative dissection of cell-to-cell communication in the developing brain.**

Automated microinjection was performed on organotypic slices of mouse E14.5–E15.5 dorsal telencephalon using a solution containing Dextran-A555 (not shown) and Alexa488 (green). Slices were fixed after microinjection (n = 71 cells total) and were stained for Tbr2 (white, H–J).

A–C Representative examples of a non-coupled cell (A, cartoon on the right), a 2-cell cluster (B, cartoon on the right), and a 5-cell cluster (C, cartoon on the right). The asterisks indicate the Dx-A555-positive, microinjected cell.

D Percentage of microinjected cells found in a coupled cluster (coupled, green).

E Cluster size, expressed as % of total.

F Distribution of coupled cells expressed as the distance from the ventricular surface (0 µm = ventricle surface).

G Distribution of coupled cells divided into 11 bins and expressed as % of total.

H, I A representative picture of (H) a 2-cell cluster (cartoon in the middle, nuclei are numbered from 1 to 2) and (I) a 7-cell cluster (cartoon in the middle, nuclei are numbered from 1 to 7), the latter containing one Tbr2-positive cell. Microinjected cells are stained for Tbr2. The asterisks indicate the Dx-A555-positive, microinjected cell.

J Comparison of Tbr2-positive cells (expressed as % of total) among all the cells in the VZ (VZ; n = 1,003 from four different confocal images from three independent experiments; error bar represents standard deviation) and in coupled clusters in the VZ (coupled clusters, VZ; n = 104 from two independent experiments).

K Coupled cells were scored based on presence/absence of apical and/or basal polarity cues, and results are expressed as % of total.

Data information: In (A–C, H, and I), the images on the left are maximum intensity projections (MIP) of 18, 17, 23, 12, and 30 focal planes, respectively; in (H, I), the images on the right are single optical sections corresponding to the nucleus (white dotted line); scale bars are 20 µm for MIP and 10 µm for single focal planes. VZ, ventricular zone; SVZ, subventricular zone; CP, cortical plate.

Source data are available online for this figure.

---

the basal surface and was programmed to target cells at a variable depth (10–35 µm) from the basal surface. The microinjections were spaced 30 µm apart to better visualize neural morphology (Fig 6B–H). As shown in Fig 6C and D, neurons in different layers, corresponding to different neuronal subtypes, were targeted (n = 133 neurons injected in total).

The subtypes targeted were Reelin-positive, Cajal–Retzius neurons (17% of cells, n = 12 cells in total; Fig 6C, bottom row and E) and Reelin-negative, newborn pyramidal neurons (83% of cells, Fig 6C, top row and E). The pyramidal identity was confirmed using a staining for the upper-layer markers SatB2 (76% of cells SatB2 positive; n = 39 cells in total; Fig 6D and F). These data demonstrate that the Autoinjector can target two neuronal subtypes present in the developing mouse telencephalon.

We then used the Autoinjector to target the developing human brain. The use of human tissue provides an invaluable tool to study neocortex development and evolution [12,15,28–34]. We first tested if the Autoinjector could target APs in organotypic slices of fetal stage 12 weeks post-conception (12 wpc) human telencephalon. We found that the Autoinjector could successfully target APs from the apical surface of a human organotypic slice (Appendix Fig S8). Additionally, the Autoinjector targeted human neurons and revealed their morphology (Fig 6G and H). Thus, the Autoinjector can be used to target the same cell types in mouse and human organotypic slices. These data strongly indicate that the Autoinjector can potentially be adapted and optimized for other organisms to study development and evolution at the cellular level.

## Discussion

Here, we report the development of the Autoinjector, an image-guided robotic device that can perform microinjection into single cells in tissue in an automated fashion.

A central goal in biology is understanding how single-cell behavior impacts on tissue development and function. In the developing brain, a dynamic and complex tissue, neural stem cells divide and give rise to neurons. Several techniques have been developed to label, manipulate, and follow single neural stem cells in the developing brains of different species

[6,32,35–46]. All techniques and systems face a trade-off between (i) the control of the composition of solution that is delivered to the cells, (ii) the number of cells that can be targeted, (iii) the time and economical effort necessary to perform it, (iv) the ability to provide single-cell resolution, and (v) the physiological relevance of the model system. Here, we show that the Autoinjector can address and overcome most of these major challenges and limitations.

Firstly, the automated microinjection into single cells provides great control over the solution delivered into the cells, both in terms of its chemical composition and its complexity, as shown by the injection of the fluorescent dye(s) (either alone or in combination) and mRNAs.

The Autoinjector enabled us to inject hundreds of APs across 1 mm of the apical surface of an organotypic slice of the mouse telencephalon, a feat previously too challenging to achieve with manual microinjection. The superior performance of the Autoinjector is linked to an increased success rate of microinjection and decreased time of the injection procedure relative to manual systems, resulting in a 3-fold to 15-fold increase in performance compared with the manual injection platform. Of note, the Autoinjector is fully open-source and can be implemented on any standard microscope setup with minor modifications and without significant additional costs. A complete parts list, the software package, and instructions for assembling the hardware have been made available for the reader (see Data and Code Availability).

The excellent single-cell resolution provided by the Autoinjector allowed us to follow the injected cells and their progeny up to 24 and 48 h in organotypic slice culture and to assess their morphology, location, and identity. The Autoinjector will allow in the future to systematically study lineage progression at specific times during development, by combining morphological assessment of single-cell behavior with the respective transcriptome profiling. This would pave the way to a better understanding of the genetic logic of cell fate specification during development and evolution.

Furthermore, we used the excellent single-cell resolution achieved with the Autoinjector to gain insight into the cell biology of neural stem cells, focusing on gap junction communication between APs in E14.5 organotypic slices of the mouse telencephalon. Gap junctions are channels allowing the exchange of

small molecules across populations of cells during development [24,47]. Although gap-junctional coupling has been studied in neural progenitors using whole-cell patch clamping [25,47], to our knowledge the stem cell type specificity of coupling has not been addressed before. We used the Autoinjector to address this point and, consistent with previous reports [25], found that coupled clusters are formed preferentially by cells that reside in the

ventricular zone. In addition, we could quantitatively assess that (i) coupled clusters have a variable size and 30% of clusters are two-cell clusters, (ii) the two-cell clusters are not daughter cell clusters, (iii) coupled clusters include Tbr2-negative and Tbr2-positive cells, suggesting the existence of clusters containing both APs (Tbr2 negative) and BPs (Tbr2 positive), and (iv) the apical attachment is a key factor influencing the ability of a cell to be

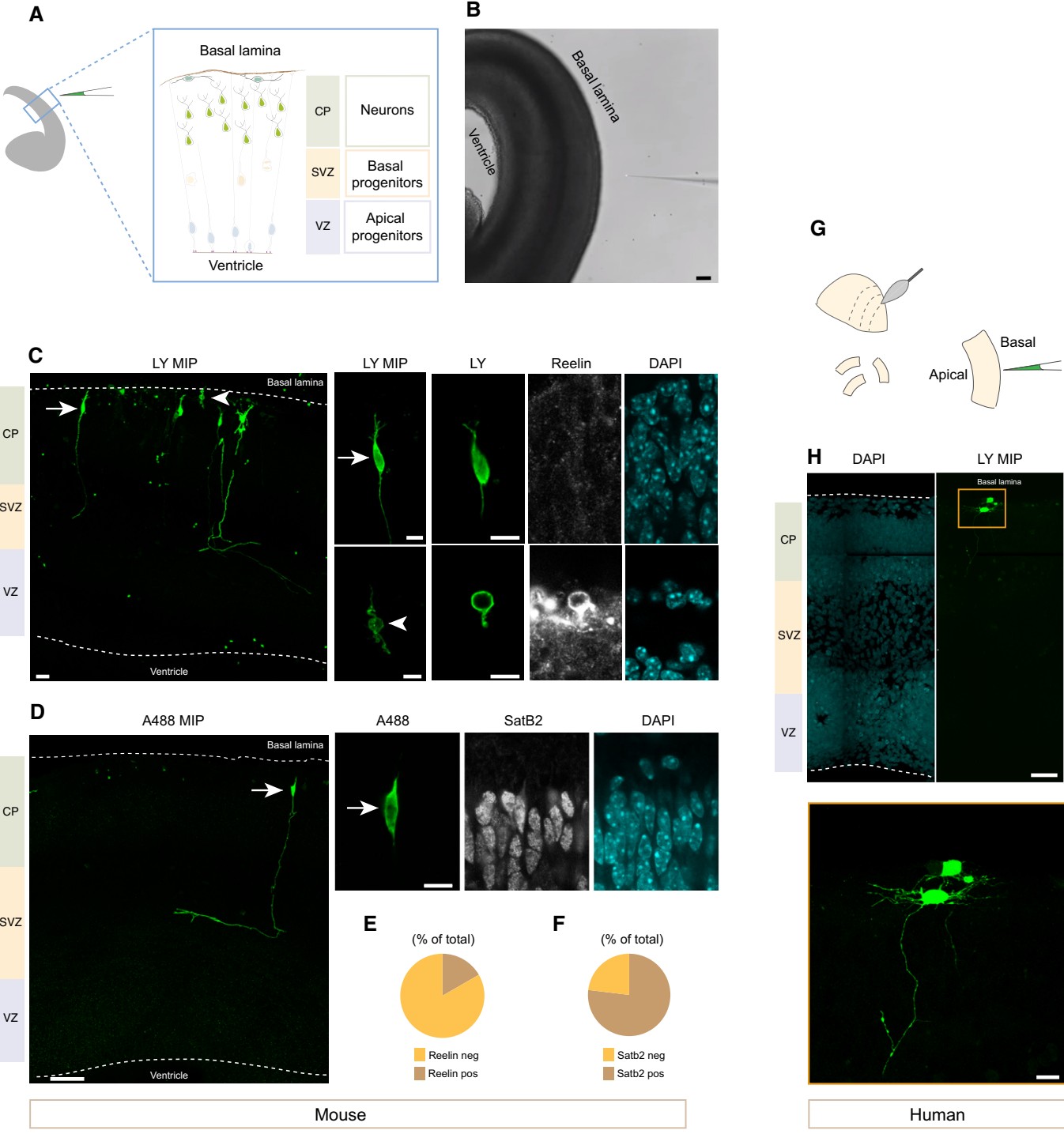

**Figure 6.**

**Figure 6.  Automated microinjection platform can be applied in a generalized manner to target neurons in the developing mouse and human telencephalon.**

Automated microinjection of mouse (A–F) and human (G, H) neurons. Automated microinjection was performed on organotypic slices of mouse E16.5 or human 12 wpc dorsal telencephalon using Dx-A555 (not shown) and LY (green, C, D, H). Slices were fixed after microinjection (C, D–F, H) and stained for Reelin (white, C) and SatB2 (white, D).

A   Schematic of automated microinjection into neurons.
B   Phase contrast image of automated microinjection into mouse neurons. Scale bar is 100 μm.
C   Left: overview image of a mouse microinjected area (arrow indicates a pyramidal neuron, arrowhead indicates a Cajal–Retzius neuron; image is MIP of 20 focal planes). Microinjected neurons show different morphology and positivity for the Cajal–Retzius marker Reelin (white). Right, top row: the pyramidal neuron indicated with an arrow in the overview image is negative for Reelin. Right, bottom row: the Cajal–Retzius neuron indicated with an arrowhead in the overview image is positive for Reelin. Images on the top and bottom row left (LY MIP) are a MIP of 2 and 3 focal planes, respectively. The three images on the top and bottom right (LY, Reelin and DAPI) are single optical sections. Scale bar is 20 μm for MIP and 10 μm for single focal planes.
D   Overview image of a mouse microinjected area. The arrow indicates a pyramidal neuron; the image is MIP of 30 focal planes, scale bar: 50 μm. Row on the right: the microinjected neuron with pyramidal morphology indicated with an arrow in the overview is positive for the upper-layer marker SatB2 (white). Images are single focal planes. Scale bar is 10 μm.
E   Proportion of neurons positive or negative for Reelin (*n* = 12 cells total).
F   Proportion of neurons positive or negative for SatB2 (*n* = 39 cells in total).
G   Schematic of human tissue hand slicing and microinjection.
H   Microinjected neurons in the developing human telencephalon. Top, MIP of 20 focal planes (scale bar: 50 μm). Bottom: high magnification (scale bar: 10 μm).

Data information: VZ, ventricular zone; SVZ, subventricular zone; CP, cortical plate.
Source data are available online for this figure.

part of a cluster. The ability of the Autoinjector to be easily adapted to different tissues and species allows for a systematic analysis of junctional coupling in different areas of the neural tube (dorsal vs. ventral telencephalon, hindbrain) and in species with different brain size.

Finally, we show that the Autoinjector can be applied in generalized manner to target other cell types and species of relevance. The precise spatial control of the Autoinjector allowed us to inject pyramidal neurons in the mouse developing telencephalon by consistently performing microinjections at a depth of 30 μm from the basal surface. By simply changing the depth of injection to 10 μm, the Autoinjector was able to target Cajal–Retzius neurons in both the mouse and human developing brain, illustrating the flexibility of our robotic device. The Autoinjector can therefore be used to manipulate and study different types of neurons in the developing brain of different species. If extended to the post-natal brain, the Autoinjector may enable studying the molecular mechanisms governing synapse formation, a crucial event involved in learning and memory [48–50] and in developmental disorders such as autism spectrum disorder (ASD) and mental retardation [51,52]. Additionally, by simply changing the developmental stage at which the injection is performed, one would be able to target any neuronal type/layer. Furthermore, other cell types may be targeted in different tissues or/and at different depths within the same tissue, provided (i) a good anatomical and histological knowledge about the tissue organization and (ii) accessibility of the cells via the tissue surface(s). Future work could focus on expanding the Autoinjector platform to inject other tissues (e.g., the developing skin), developmental model organisms (chick, zebrafish, *Drosophila*, *Caenorhabditis elegans*), and even non-model organisms for which transgenesis is not available.

In conclusion, we demonstrated that the combination of robotic automation, real-time image acquisition and analysis leads to a significant increase in the efficiency of a difficult laboratory technique, such as microinjection into single cells in tissue. Robotic systems have enabled the automation of difficult laboratory techniques that require precise micromanipulation such as *in vivo* patch clamping of single [53–55] as well as multiple neurons *in vivo* [56]. Additionally, previous work relied on camera images to guide automated patch clamping systems to specific locations in tissue [11,57]. These applications resulted in significant improvement in the success of patch clamping and enabled neuroscientists to perform complex experiments previously limited by technical difficulties. To our knowledge, this is the first attempt to automatically perform microinjection into single neural stem cells and newborn neurons in organotypic brain slices. The principle we developed of using visual cues to target specific locations can be applied to any tissue with an *a priori* knowledge of the location of cells. Based on the high efficiency we achieved in injecting APs and newborn neurons both in the mouse and in the human telencephalon, we predict that this process will be further implemented in applications where microinjection was previously not considered possible.

# Materials and Methods

### Microinjection hardware

We designed the Autoinjector (Fig 1) by modifying a standard microinjection system described previously [5]. The Autoinjector hardware is composed of a pipette mounted in a pipette holder (64-2354 MP-s12u, Warner Instruments, LLC) attached to a three-axis manipulator (three-axis uMP, Sensapex Inc) for precise position control of the injection micropipette. A microscope camera (ORCA, Hamamatsu Photonics) was used for visualizing and guiding the microinjection, and a custom pressure regulation system adapted from previous work [53] was built for programmatic control of injection pressure. The pressure regulation system consisted of manual pressure regulator (0–60 PSI 41795K3, McMaster-Carr) that downregulated pressure from standard house pressure (~ 2,400 mbar) to 340 mbar. The output from the manual pressure regulator was routed to an electronic pressure regulator (990-005101-002, Parker Hannifin) that allowed fine tuning of the final pressure going to the injection micropipette (0–250 mbar) using the control software. A solenoid valve (LHDA0533215H-A, Lee Company) was then used to digitally switch the pressure output to the injection micropipette. A microcontroller (Arduino Due, Arduino) was used to control electronic pressure regulation via a 0–5 V analog voltage signal and the

solenoid via a digital transistor transistor logic (TTL) signal (Fig 1A and C). The computer controlled the three-axis manipulator via an Ethernet connection and controlled the camera and microcontroller via universal serial bus (USB) connections. All hardware was controlled by custom software as described in the next section (see User Manual for additional information about hardware).

### Microinjection software and operation

All software was written in python (Python Software Foundation) and Arduino (Arduino) and is available for download with instructions at https://github.com/bsbrl/autoinjector. We developed a graphical user interface (GUI) in python to operate the microinjection platform (Appendix Fig S1). The GUI allowed the user to image the tissue and micropipette and to customize the trajectory of microinjection (see User Manual for additional information about software).

### Mouse slice preparation

All animal studies were conducted in accordance with German animal welfare legislation, and the necessary licenses were obtained from the regional Ethical Commission for Animal Experimentation of Dresden, Germany (Tierversuchskommission, Landesdirektion Dresden). Organotypic slices were prepared from E14.5 or E16.5 mouse embryonic telencephalon or hindbrain as previously described [6]. C57BL/6 mouse embryos were used (Janvier Labs). Briefly, the mouse telencephalon was dissected at room temperature in Tyrode's solution. After the removal of meninges, the tissue was embedded 3% low-melting agarose (Agarose Wilde Range, A2790; Sigma-Aldrich) in PBS at 37°C. After solidification of the agarose upon cooling to room temperature, 300–400 μm coronal slices were cut using a vibratome (Leica VT1000S; Leica). The slices were transferred to 3.5-cm dishes containing 37°C warm slice culture medium [SCM: Neurobasal medium (Thermo Fisher Scientific), 10% rat serum (Charles River Japan), 2 mM L-glutamine (Thermo Fisher Scientific), Penstrep (Thermo Fisher Scientific), N2 supplement (17502048; Thermo Fisher Scientific), B27 supplement (17504044; Thermo Fisher Scientific), 10 mM Hepes-NaOH pH 7.3]. Until the start of microinjection, slices were kept for 5–10 min in a slice culture incubator maintained at 37°C and gassed with a humidified atmosphere of 40% $O_2$/5% $CO_2$/55% $N_2$ (Air Liquide).

### Human slice preparation

Human fetal brain tissue was obtained from the Klinik und Poliklinik für Frauenheilkunde und Geburtshilfe, Universitätsklinikum Carl Gustav Carus of the Technische Universität Dresden, with informed written maternal consent, followed by elective pregnancy termination. Research with human tissue was approved by the Ethical Review Committee of the Universitätsklinikum Carl Gustav Carus of the Technische Universität Dresden (reference number: EK100052004). Immediately after termination of pregnancy, the tissue was placed on ice and transported to the laboratory. The sample was transferred to ice-cold Tyrode's solution, and tissue fragments of cerebral cortex were identified and dissected [58]. The tissue fragment suitable for microinjection was kept in SCM in a humidified and oxygenated bottle at 37°C for at least 1 h to allow the tissue to recover. Organotypic slices were then cut by hand

using a micro-knife. The thickness of the slices was variable and ranged between 300 and 500 μm. Slices were transferred to a 3.5-cm dish containing 37°C warm SCM and kept in the slice culture incubator until the start of microinjection.

### Micropipette preparation

Glass capillaries [1.2 mm O.D. × 0.94 mm I.D., Harvard Apparatus (BF-120-94-10)] were pulled into micropipettes using a micropipette puller (P-97 Flaming Brown, Sutter instruments, Novato, CA). See User Manual and guidelines therein.

### Microinjection solution preparation

All the injection solutions were made up with RNase-free bidistilled water. Solutions always contained a fluorescently labeled dye (Alexa-coupled Dextrans and/or Lucifer Yellow; Thermo Fisher Scientific, from 10 μg/μl stocks (for the complete list of fluorescently labeled dyes used, see Appendix Table S1) at 2–5 μg/μl to trace the microinjected cells and their progeny. For the mRNA injections, *in vitro*-transcribed (ivt) poly-A mRNA for RFP was prepared as previously described [6] using the T7 mMessage-mMachine Ultra kit (Thermo Fisher Scientific). Briefly, RFP ivt-RNA was dissolved in RNase-free bidistilled water at 1 μg/μl and snap-frozen as 1–2 μl aliquots. Before the start of microinjection, RNA was heated to 90°C for 45 s and cooled to 4°C before the addition of the Dextran-Alexa 488. For the coupling experiments, Dextran-Alexa555 was mixed with the Alexa488, whose molecular weight (MW: 884.91 Da) is below the reported cut-off of the gap-junctional channels [59]. For the neuron injections, Dextran-A555 was mixed with Lucifer Yellow (LY), a low molecular weight fluorescent tracer allowing for a better visualization of the neuronal morphology [60]. All solutions for microinjection were centrifuged at 16,000 $g$ for 30 min at 4°C. The supernatant was collected, kept on ice, and used for microinjection. The injection solution was loaded into micropipettes, and the micropipette was mounted onto the micropipette holder.

### Microinjection

For additional information on how to run the Autoinjector, see User Manual and Movies EV4–EV7. Immediately before the start of microinjection, brain slices were transferred to 3.5-cm dishes containing 37°C warm $CO_2$-independent microinjection medium [CIMM: DMEM-F12 (Sigma D2906) 2 mM L-glutamine, Penstrep, N2 and B27 supplements, 25 mM (final concentration) Hepes-NaOH pH 7.3] and positioned on the microscope stage (Axiovert 200; Zeiss, Jena). The micropipette was brought into the field of view of the microscope close to the edge of the tissue using a 10× objective lens (Zeiss, Jena). To verify that the micropipette was not clogged, pressure was applied through the GUI, and the appearance of a fluorescent signal was checked. If no fluorescence was observed, the micropipette was changed and the positioning was repeated. If the micropipette was in optimal conditions (not clogged), the microinjection was carried out using a 20× objective lens (Zeiss, Jena) to allow for a better visualization of the procedure. Manual microinjections were performed as described previously [5]. Automated microinjection was performed as illustrated in Movies EV1 and EV2.

## Slice culture

Slices were prepared as previously described [6,61]. The slices were immersed in collagen and transferred with ~ 200 µl of collagen solution into the 14-mm well of a 35-mm Glass Bottom Microwell Dish (MatTek). The dish was placed for 5 min on a heating block at 37°C. This transfer was defined as $t = 0$ of slice culture and was followed by 40 min in the slice culture incubator to allow the collagen to solidify. The dish then received 2 ml of SCM, and slice culture was continued in the slice culture incubator for the indicated times (24 or 48 h). For the EdU incorporation experiments, EdU (final concentration 2.5 µg/ml) was added to both collagen and SCM.

## Tissue fixation, fluorescence staining and imaging

At the end of the slice culture, the SCM was removed, the dish was rinsed twice with fresh PBS, and the slices were fixed in 4% paraformaldehyde in 120 mM sodium phosphate buffer pH 7.4 at room temperature for 30 min followed by 4°C overnight. Slices were removed from the collagen matrix using forceps and extensively washed in PBS before re-embedding in 3% low-melting agarose (Carl Roth) in PBS. After agarose solidification, 50-µm vibratome sections (also referred as floating sections) were cut parallel to the original cutting plane. Floating sections were collected in a 24-multiwell and processed for immunofluorescence using standard procedures (for the complete list of primary and secondary antibodies used, see Appendix Table S1). Floating sections were stained with 4′,6-diamidino-2-phenylindole (DAPI, Sigma-Aldrich) and mounted on a glass slide in Mowiol 4-88. EdU was detected using the Click-It kit (Molecular Probes). Samples were analyzed by confocal microscopy (Zeiss LSM 780 NLO; Zeiss, Jena). Unless indicated otherwise, all images shown are 0.8-µm-thick single optical sections.

## Image analysis

All images were analyzed using ImageJ [62] and FiJI [63]. The cell's contours were traced using the fluorescent microinjection dye. Cell contours were stored using the ROI (region of interest) manager and were used (i) to identify the nucleus (ii) to determine the presence of Tbr2, TuJ1, and/or Reelin expression.

For the distance calculation, we calculated the distance between the ventricular surface (defined as 0 µm) and the center of the nucleus (identified in the optical section corresponding to the largest nuclear area). The distance values were binned as follows: 0 to < 20 µm, 20 to < 40 µm, 40 to < 60 µm, 60 to < 80 µm, 80 to < 100 µm, 100 to < 120 µm, 120 to < 140 µm, 140 to < 160 µm, and > 160 µm.

## Statistical analysis methods

All values reported represent the mean and ± represents standard deviation from the mean calculated using Microsoft Excel. Data were tested for normal distribution using a one-sample Kolmogorov–Smirnov test to determine the appropriate statistical test to use in MATLAB® (stat_test_figure3.m, stat_test_appendixnote3.m, stat_test_appendixfigureS4.m, stat_test_appendixfigureS5.m). These results found that the data did not originate from a normal distribution; thus, a Mann–Whitney $U$-test (also called Wilcoxon rank-sum test) was used. Testing for significance was done using a one-tailed (stat_test_figure3.m, stat_test_appendixfigureS4.m, stat_test_appendixfigureS5.m), or a double-tailed (stat_test_appendixnote3.m) Mann–Whitney $U$-test using MATLAB®. A $P$-value < 0.05 was considered significant with $P < 0.05$ denoted with *, $P < 0.001$ denoted with **, and $P < 0.0001$ denoted with ***.

## Data and code availability

All numerical data have been provided as source data for every figure. Image count data contained in this document are based on confocal images which are available upon request. Code developed for establishing and running the Autoinjector can be found at https://github.com/bsbrl/autoinjector. In addition, the Sensapex ump micromanipulator dll file is required for operation and may be available from Sensapex upon request.

Expanded View for this article is available online.

## Acknowledgements

We are grateful to the Services and Facilities of the Max Planck Institute of Molecular Cell Biology and Genetics for the outstanding support provided, notably J. Helppi and his team of the Animal Facility, J. Peychl and his team of the Light Microscopy Facility, and Jan Wagner and the MPI-CBG mechanical workshop. We would like to thank Prof. Dr. med. Pauline Wimberger and Dr. med. Nannette Grübling from the Klinik und Poliklinik für Frauenheilkunde und Geburtshilfe, Universitätsklinikum Carl Gustav Carus of the Technische Universität Dresden, for providing fetal human tissue, and Michael Heide and Katie Long for dissecting and preparing the human fetal brain tissue. We would like to thank the Dr. Svante Pääbo's entire team for helpful discussions, and Mareike Albert, Katie Long, and Nicola Maghelli for critical reading of the article and for their helpful comments. SBK acknowledges funds from the Mechanical Engineering department, College of Science and Engineering, MnDRIVE RSAM initiative of the University of Minnesota, McGovern Institute Neurotechnology (MINT) fund, National Institutes of Health (NIH) 1R21NS103098-01. GS was supported by NSF IGERT training fellowship, and the NSF IGERT travel award. ET was supported by NOMIS Distinguished Scientist Award for Dr. Svante Pääbo. WBH was supported by grants from the DFG (SFB 655, A2), the ERC (250197), and ERA-NET NEURON (MicroKin).

## Author contributions

SBK, ET, GS, and WBH conceptualized the technology. GS wrote all Python and Arduino code. SBK, ET, GS, and CH conducted experiments. SBK, ET, GS, and CH analyzed the data. SBK, ET, GS, CH, and WBH wrote the article.

## Conflict of interest

The authors declare that they have no conflict of interest.

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
