## [Review Process File · EMBO Reports]

Robotic platform for microinjection into single cells in brain tissue

Gabriella Shull, Christiane Haffner, Wieland B. Huttner, Suhasa B. Kodandaramaiah and Elena Taverna

Review timeline:	Submission date:	6 February 2019
	Editorial Decision:	6 March 2019
	Revision received:	16 June 2019
	Editorial Decision:	8 July 2019
	Revision received:	23 July 2019
	Accepted:	7 August 2019

Editor: Esther Schnapp

Transaction Report:

1st Editorial Decision

6 March 2019

Thank you for the submission of your manuscript to EMBO reports. We have now received the full set of referee reports that is pasted below.

As you will see, the referees find the automated microinjection tool you describe potentially interesting and useful. However, they also all note that the study (and the data) would need to be strengthened before it can be considered for publication here. Upon cross-commenting on each others' reports it became clear that referee 2's request for data on other tissues and model organisms may be out of the scope of this study and can be overruled. Instead, testing different molecules and/or substances and compare their efficiencies is a point both referees 2 and 3 agree on and it should therefore be addressed.

Given the overall constructive comments, we would thus like to invite you to revise your manuscript with the understanding that the referee concerns must be fully addressed and their suggestions taken on board. Please address all referee concerns in a complete point-by-point response. Acceptance of the manuscript will depend on a positive outcome of a second round of review. It is EMBO reports policy to allow a single round of revision only and acceptance or rejection of the manuscript will therefore depend on the completeness of your responses included in the next, final version of the manuscript.

Revised manuscripts should be submitted within three months of a request for revision; they will otherwise be treated as new submissions. Please contact us if a 3-months time frame is not sufficient for the revisions so that we can discuss this further. Given your 6 main figures, I suggest that you layout your manuscript as a full research article.

Supplementary figures, tables and movies can be provided as Expanded View (EV) files, and we can offer a maximum of 5 EV figures per manuscript. EV figures are embedded in the main manuscript text and expand when clicked in the html version. Additional supplementary figures will need to be included in an Appendix file. Tables can either be provided as regular tables, as EV tables or as

Datasets. Please see our guide to authors for more information.

Regarding data quantification, please specify the number "n" for how many independent experiments were performed, the bars and error bars (e.g. SEM, SD) and the test used to calculate p-values in the respective figure legends. This information must be provided in the figure legends. Please also include scale bars in all microscopy images.

We now strongly encourage the publication of original source data with the aim of making primary data more accessible and transparent to the reader. The source data will be published in a separate source data file online along with the accepted manuscript and will be linked to the relevant figure. If you would like to use this opportunity, please submit the source data (for example scans of entire gels or blots, data points of graphs in an excel sheet, additional images, etc.) of your key experiments together with the revised manuscript. Please include size markers for scans of entire gels, label the scans with figure and panel number, and send one PDF file per figure.

- a complete author checklist, which you can download from our author guidelines (<http://embor.embopress.org/authorguide#revision>). Please insert page numbers in the checklist to indicate where in the manuscript the requested information can be found. The completed author checklist will also be part of the RPF (see below).
- a letter detailing your responses to the referee comments in Word format (.doc)
- a Microsoft Word file (.doc) of the revised manuscript text
- editable TIFF or EPS-formatted figure files in high resolution. In order to avoid delays later in the process, please read our figure guidelines before preparing your manuscript figures at: http://www.embopress.org/sites/default/files/EMBOPress_Figure_Guidelines_061115.pdf

I look forward to seeing a revised version of your manuscript when it is ready. Please let me know if you have questions or comments regarding the revision.

REFeree REPORTS

Referee #1:

Shull and colleagues present an interesting and useful resource for microinjection of femtoliter volumes of liquids into single cells. The authors construct a relatively simple device and provide open source code for access to a wider community. The authors demonstrate increased efficiency of their automated approach when compared to delicate manual microinjection. Next the authors show that besides dyes exogenous mRNA can be delivered for putative analysis of candidate genes. The authors utilize their method to study the level of connectivity via gap junctions between apical progenitors in the developing mouse brain, and present interesting results. Lastly, they show that their method is also applicable to slices of human tissue and that postmitotic cells can be injected to study for instance neuronal morphology.

Overall the resource presented by Shull and colleagues has the potential to be useful for a wider community of neurobiologists, developmental- and stem cell biologists. The authors demonstrate

convincingly the utility of their device and approach, and present some novel insights relevant for neuronal progenitor stem cell biology and specifically regarding the communication via gap junctions. The manuscript is written well and the data illustrated nicely. I have some comments that however need some attention.

1. The first sentence of the abstract is too long and not easy to comprehend.
2. Figure 2 should be optimized in the following ways: apical/basal and dorsal/ventral should be indicated in all relevant images/schematics. The orientation of the slice in B/C is opposite to the schematic in A. This is confusing for the non-specialist. Panel B requires a scale bar.
3. Figure 3: 'Pressure'/'Depth' and 'Dextran' labels are very small and should be written using bigger font. Scale bars are too small and should be adjusted. There is no illustration of a DAPI-labeled slice as described in the results section. Please add one example and ideally at different resolutions to illustrate these data.
4. Figure 4. Similar like in Figure 3, the fonts labeling 'Cell 1', 'Cell 2', 'Daughter 1' etc. are too small and hard to read. While Tbr2 and TuJ1 labeling is informative, the authors should also show co-staining with an apical progenitor marker, such as PAX6 or SOX2. According to their data, the fraction of Dx/PAX6 or SOX2 double positive cells should be around 90%?
5. Figure 4H, the authors should critically discuss the fact that (only) ~47% of cells express RFP after 24h. A discussion about this feature is particularly important since the authors seem to wish advertising their method for the study of candidate genes. Such information/discussion is important so that the user may carry out appropriate controls by using different fluorophores in distinct experimental paradigms. A few sentences in the discussion may be useful to also elaborate about this property in comparison with other methods, especially genetic ones.
6. At certain passages the discussion seems slightly biased and I suggest a few revisions:
 - I) The authors state: 'Several techniques have been developed to label, manipulate and follow single neural stem cells in the developing brains of different species.' This is fine and a number of important citations have been added but the authors mainly cite their own studies and the work from the Kriegstein lab. This is necessary and fine but I suggest adding also some citations where people use genetic methods such as Cre/LoxP-mediated recombination and/or CRISPR-based approaches like the recently described SLENDR method to achieve single cell labeling and manipulation.
 - II) At the end of the first paragraph, the authors state that 'the Autoinjector can address all these major challenges and limitations.' In my opinion this claim goes a bit too far. I suggest replacing 'all' with 'most', 'many', or 'a number of'.
 - III) On page 16 the authors state: '...to systematically study lineage progression by combining...'. Again this claim goes a bit too far because the authors inject dye at E14.5 and analyze the samples after 24h or 48h. Even if the dye is injected earlier, this approach can at best be used to study lineage progression in a (short) defined time window. While still potentially giving very useful and interesting results the authors may want to rephrase the relevant sentence(s).
 - IV) Page 17, the authors mention: 'We can anticipate that the autoinjector will allow the dissection of the molecular mechanisms governing synapse formation,...'. I am not sure that this will be easily possible since the authors study the developing brain during embryogenesis and in slice culture conditions. Most of the neurons are only born/migrating during this time and very few form synapses (which also are probably very different from the synapses in a mature brain). I also think that the link to diseases is not justified in this regard. I suggest revising this section accordingly.
7. The title for the figure legend for Figure 4 should be rephrased since there is no genetic manipulation shown in the actual figure.

Referee #2:

This manuscript by Shull, Haffner et al. describes a new tool for the microinjection of single cells in brain slices. Using a robotic platform instead of performing the injections manually, they achieve high throughput, high efficiency and reduce the need for lengthy training periods for the user. They

then go on to demonstrate that the technique is suitable for tracing the lineage progression of progenitor cells, express exogenous mRNAs, assess the direct cell-cell communication between progenitors and target different cell types in the developing brain. The robotic platform is undoubtedly a significant improvement that will make microinjection feasible and available to the developmental neurobiology community. However, further demonstration of the versatility of the technique needs to be provided in order to justify the claims of wide applicability for other fields. Similarly, some of the results presented appear still preliminary and further experiments are needed before conclusions can be drawn.

Major comments:

-Throughout the manuscript, including the title, the authors refer to organotypic slices as "intact brain tissue". This is highly misleading and the fact that the whole manuscript is based on the use of organotypic cultures (ex vivo explants made by slicing brain tissue) should be made clearer, particularly in the title and abstract.

-The impact of multiple microinjections on cell and tissue viability is not assessed. The amount of cell death occurring due to the procedure should be quantified and documented. A comparison to the cell death rate when doing the microinjections manually should be included, not only for the injected cells but also for neighboring cells (this is particularly important when injections are performed deeper into the tissue). Similarly, a comparison of cell viability when delivering different "substances" is missing (e.g.: delivery of a dye versus mRNA).

-Despite the potential for high throughput with the robotic platform, "n" numbers remain very low for most experiments. For instance, for mRNA injection (4H), 17 cells and for the quantification of reelin+ cells (6D), 12 cells were considered. It is also not clear how many slices were used in each experiment nor the reproducibility of the results with the use of more biological replicates.

-Further evidence is needed to support the claim that cell coupling is linked to the presence of apical attachment of the cells. As currently presented, it rather looks like coupling is highly correlated with targeting cells during the mitotic phase (which occurs apically) and could therefore reflect targeting two daughter cells in the process of separating. In fact, since apical progenitors in the VZ constitute a pseudostratified epithelium, it should be possible to target them when their cell bodies are away from the VZ (not in M-phase) and still visualize their connections.

-The claim of high versatility of the technique should be backed up by data showing its usefulness for targeting adult brain and different types of tissue (other than neural), and at least one other species (zebrafish, chick, drosophila or c.elegans as suggested by the authors). Same applies to the versatility of substances that can be delivered. Can proteins or drugs be also microinjected successfully with the technique?

Minor comments:

-It is not currently clear in the text how much time does the optimization of the parameters take, especially for an untrained user. In this same line, does the optimal pressure depend on the type of cell or type of tissue to be microinjected? How thick should the injection capillaries be and does this, again, depend on cell type/tissue? A list of general guidelines or a table with the parameters used for different cells/substances, would be very useful.

-A high magnification image of a single cell being injected would really help having a sense of the invasiveness of the technique.

-When assessing the coupling between cells (page 12), it is stated that the tissue is fixed immediately after microinjection. The timing should be given in a more specific manner as the average time from injection to fixation (the first cells injected had longer periods until fixation).

-The authors show nicely how the lineage progression of the microinjected cells in the slices can be followed over time. How does this compare to the timing of lineage progression of progenitors in the intact brain?

-Can the robotic platform aid the injection of multiple substances in the same slice? And do the authors envision a possibility of eventually targeting single cells in living organisms (not necessarily

mice)?

-The text in the abstract needs to be revised, as well.

Referee #3:

The manuscript by Shull et al described an automated microinjection platform, called Autoinjector, that consists in using images acquired by a microscope to guide a microinjection pipette. This customized platform allows significant gain of time and precision, with an increased number of cells successfully injected within a shorter period. In addition, while manual injection needs a lot of practice to be mastered, Autoinjector can be operated by an inexperienced user. Although this platform brings many advantages and although authors provide a user manual, implementation of the technology in other laboratories may be limited to qualified users (expertise in microscopy as it requires modification of the standard set-up). In addition, to better assess the performance of the Autoinjector, authors should address the following points :

- 1/ Authors have tested several depths for microinjection in APs (ranging from 10 to 35 μm). In fig.3A, Authors show a 35% success rate with the following parameters: depth 10 μm ; pressure: 75mbar. In fig.3B and 3C, they show a rate of 68% and 43%, respectively, using the exact same conditions. Could the authors comment on these discrepancies? Have the authors really assess the variability of the platform (from one day to the other, etc...)?
- 2/By injecting the APs, one would expect labelling of the whole cell including the glial process that extend from the ventricle to the pia. Authors do show such a staining but only in one cell (fig3B). Why processes are not labelled in fig. 3A and B?
- 3/One advantage of microinjection is injection of non-charged molecule. It would have been really valuable to show that Autoinjector also increases the rate of success of injection with such molecule.
- 4/Authors claim that injection of a gap junction permeable dye might help defining how many cells are coupled via gap junction. Could the volume of injected dye influence the results? It is quite puzzling to see similar fluorescence intensity in cells in small and large clusters (fig5a versus 5C). If the same volume has been injected, we would expect a decrease of the signal in the large cluster.
- 5/Quid of cell death over time? Does use of Autoinjector affect survival in the same way than manual injection?
- 6/ Authors are used 200 to 500 μM -thick organotypic section. Is the thickness of the section influencing the yield of injection?
- 7/ Fig4C: not clear if Daughter cells 3 and 4 are negative for both Tbr2 and Tbr1. If so, what is the fate of those cells, APs? Their localization argue against this hypothesis. Could the authors comment? In fig4, it would have been interesting to provide a sax2 or Pax6 staining. On the same line, in fig 6C, an upper layer (cuX1) staining is missing.
- 8/ As the yield of injection is pretty low in hindbrain and human APs, what is the added value of Autoinjector?
- 9/ When performing injection from the basal surface, do the author target APs (through basal feet of the radial process)? At which rate?

Minor points:

- First sentence of the abstract may be revised.
- reference to fig 5G and I are missing.
- page 12: should be 5J instead of 5K
- page 13: should be 5K instead of 5J
- discussion page 15: why authors are mentioning injection across 1mm while they performed injection with depth ranging from 10 to 25 μm (for APs)?

1st Revision - authors' response

16 June 2019

Response to Reviewers

Reviewer #1

Reviewer's Comment:

Shull and colleagues present an interesting and useful resource for microinjection of femtoliter volumes of liquids into single cells. The authors construct a relatively simple device and provide open source code for access to a wider community. The authors demonstrate increased efficiency of their automated approach when compared to delicate manual microinjection. Next the authors show that besides dyes exogenous mRNA can be delivered for putative analysis of candidate genes. The authors utilize their method to study the level of connectivity via gap junctions between apical progenitors in the developing mouse brain, and present interesting results. Lastly, they show that their method is also applicable to slices of human tissue and that postmitotic cells can be injected to study for instance neuronal morphology.

Overall the resource presented by Shull and colleagues has the potential to be useful for a wider community of neurobiologists, developmental- and stem cell biologists. The authors demonstrate convincingly the utility of their device and approach, and present some novel insights relevant for neuronal progenitor stem cell biology and specifically regarding the communication via gap junctions. The manuscript is written well and the data illustrated nicely.

We thank the reviewer for the encouraging comments.

I have some comments that however need some attention.

The first sentence of the abstract is too long and not easy to comprehend.

We have edited the first sentence to make it more readable.

Figure 2 should be optimized in the following ways: apical/basal and dorsal/ventral should be indicated in all relevant images/schematics. The orientation of the slice in B/C is opposite to the schematic in A. This is confusing for the non-specialist.

We have modified **Figure 2** to ensure the slices are oriented in the same way in all the sub figures.

Panel B requires a scale bar.

We have included a scale for in **Figure 2B**.

Figure 3: 'Pressure'/'Depth' and 'Dextran' labels are very small and should be written using bigger font. Scale bars are too small and should be adjusted.

We have now modified **Figure 3** to make the text labels larger. For consistency, we have edited all figures (**Figures 1- 6**) to ensure text and annotations are clearly visible.

There is no illustration of a DAPI-labeled slice as described in the results section. Please add one example and ideally at different resolutions to illustrate these data.

We have now included a new figure (**Appendix Figure S3**) illustrating A488 and DAPI images for cells shown in **Figure 3F**. We have also added a confocal stack (**Appendix Video S3**) showing a microinjected cell (DxA555, magenta) in a slice stained with DAPI (cyan) which illustrates our scoring methodology.

Similar like in Figure 3, the fonts labeling 'Cell 1', 'Cell 2', 'Daughter 1' etc. are too small and hard to read.

We have now modified **Figure 4** to make the labels larger and easier to read.

While Tbr2 and TuJ1 labeling is informative, the authors should also show co-staining with an apical progenitor marker, such as PAX6 or SOX2. According to their data, the fraction of Dx/PAX6 or SOX2 double positive cells should be around 90%?

We have performed new experiments to quantified Sox2 positivity of microinjected cells and their progeny at 0h, 24h and 48h. These new data along with the quantification, are included in Extended Figure 2 in the revised manuscript.

Figure 4H, the authors should critically discuss the fact that (only) ~47% of cells express RFP after 24h. A discussion about this feature is particularly important since the authors seem to wish advertising their method for the study of candidate genes. Such information/discussion is important so that the user may carry out appropriate controls by using different fluorophores in distinct experimental paradigms. A few sentences in the discussion may be useful to also elaborate about this property in comparison with other methods, especially genetic ones.

We would like to thank reviewer 1 for raising this point.

In the current study, we have achieved an mRNA translation rate ranging from 30 (hindbrain) to 50% (telencephalon) in all species analyzed (mouse, ferret and human). This is consistent with our previous work (See Taverna et al, 2012; Wong et al, 2014). While we do not have an explanation for why translation occurs in only a fraction of the cells, we can rule out trivial explanations, such as mRNA degradation during microinjection (Taverna, unpublished data).

The presence of non-translation and translating cells in the same slice offers a very good internal control for functional manipulation experiments. Indeed, one expects a mRNA-induced phenotype to become evident only in translating cells (RFP positive), and not to be present in non-translating cells (RFP negative).

We fully understand that this point is crucial for other users wishing to use our platform for functional manipulation. We have added a guidelines section to the user manual, in which we discuss this property, its significance and how to plan appropriate controls.

At certain passages the discussion seems slightly biased and I suggest a few revisions:

I) The authors state: 'Several techniques have been developed to label, manipulate and follow single neural stem cells in the developing brains of different species.' This is fine and a number of important citations have been added but the authors mainly cite their own studies and the work from the Kriegstein lab. This is necessary and fine but I suggest adding also some citations where people use genetic methods such as Cre/LoxP-mediated recombination and/or CRISPR-based approaches like the recently described SLENDR method to achieve single cell labeling and manipulation.

The revised manuscript now cites the following papers that describe the development and applications of the SLENDR method:

- Mikuni et al., 2016 (doi: 10.1016/j.cell.2016.04.044)
- Nishiyama, 2019 (doi: 10.1016/j.neures.2018.07.003)
- Mikuni, 2019 (doi: 10.1016/j.neures.2019.04.007)
- Nishiyama et al, 2017 (doi: 10.1016/j.neuron.2017.10.004)

At the end of the first paragraph, the authors state that 'the Autoinjector can address all these major challenges and limitations.' In my opinion this claim goes a bit too far. I suggest replacing 'all' with 'most', 'many', or 'a number of'.

We have toned down the language in the sentence from “Here we show that the Autoinjector can address and overcome **all** of these major challenges and limitations.”

to: “Here we show that the Autoinjector can address and overcome **most** of these major challenges and limitations.”

On page 16 the authors state: '...to systematically study lineage progression by combining...'. Again this claim goes a bit too far because the authors inject dye at E14.5 and analyze the samples after 24h or 48h. Even if the dye is injected earlier, this approach can at best be used to study lineage progression in a (short) defined time window. While still potentially giving very useful and interesting results the authors may want to rephrase the relevant sentence(s).

We have changed the sentence to indicate that lineage progression can be systematically studied at specific times during development. The sentence now says “... to systematically study lineage progression at specific times during development.”

Page 17, the authors mention: 'We can anticipate that the autoinjector will allow the dissection of the molecular mechanisms governing synapse formation,...'. I am not sure that this will be easily possible since the authors study the developing brain during embryogenesis and in slice culture conditions. Most of the neurons are only born/migrating during this time and very few form synapses (which also are probably very different from the synapses in a mature brain). I also think that the link to diseases is not justified in this regard. I suggest revising this section accordingly.

We have now modified this sentence to “If extended to the post-natal brain, the Autoinjection may enable studying the molecular mechanisms governing synapse formation”

The title for the figure legend for Figure 4 should be rephrased since there is no genetic manipulation shown in the actual figure.

We have changed the title of **Figure 4** to “Neural stem cell lineage tracing and expression of

exogenous mRNA”

Reviewer #2

This manuscript by Shull, Haffner et al. describes a new tool for the microinjection of single cells in brain slices. Using a robotic platform instead of performing the injections manually, they achieve high throughput, high efficiency and reduce the need for lengthy training periods for the user. They then go on to demonstrate that the technique is suitable for tracing the lineage progression of progenitor cells, express exogenous mRNAs, assess the direct cell-cell communication between progenitors and target different cell types in the developing brain. The robotic platform is undoubtedly a significant improvement that will make microinjection feasible and available to the developmental neurobiology community. However, further demonstration of the versatility of the technique needs to be provided in order to justify the claims of wide applicability for other fields. Similarly, some of the results presented appear still preliminary and further experiments are needed before conclusions can be drawn.

We thank the reviewer for the encouraging comments and hope the revisions made in response to the comments below satisfy the reviewer.

Major comments:

-Throughout the manuscript, including the title, the authors refer to organotypic slices as "intact brain tissue". This is highly misleading and the fact that the whole manuscript is based on the use of organotypic cultures (ex vivo explants made by slicing brain tissue) should be made clearer, particularly in the title and abstract.

In response to this comment, we have now replaced all instances of “intact tissue” with “organotypic” or “ex vivo”.

The impact of multiple microinjections on cell and tissue viability is not assessed. The amount of cell death occurring due to the procedure should be quantified and documented.

Following the reviewer’s comment, we quantified the number of picnotic nuclei in

- in vivo
- non injected slices (24h culture)
- non-injected cells in the microinjected-area in a slice injected with dye (24h culture; manual injection)
- non-injected cells in the microinjected-area in a slice injected with dye (24h culture; automated injection)
- non-injected cells in the microinjected-area in a slice injected with dye and mRNA for RFP (24h culture; automated injection)

The results are shown in **Appendix Figure S4**.

A comparison to the cell death rate when doing the microinjections manually should be included, not only for the injected cells but also for neighboring cells (this is particularly important when injections are performed deeper into the tissue). Similarly, a comparison of cell viability when delivering different "substances" is missing (e.g.: delivery of a dye versus mRNA).

It is not possible to assess cell death after microinjection in a faithful way after 24h in culture. Typically, damage caused by microinjection results in cell death within 2-3 hours. If cells manage to overcome this “critical phase” then their behavior and survival is comparable to that of nonmanipulated cells (these info is difficult to find in the primary literature; see EMBO Practical Course on Microinjection Manual).

We decided to use EdU incorporation to better assess the impact of microinjection on cell viability, for both microinjected and non-microinjected cells. We performed and quantified EdU incorporation for 24h in:

- non injected slices slice
- microinjected cells in a microinjected slice (manual injection)
- microinjected cells in a microinjected slice (automated injection)
- non-injected cells in the microinjected-area in an injected slice (manual injection)
- non-injected cells in the microinjected-area in an injected slice (automated injection)

The results are shown in **Appendix Figure S5**.

The EdU incorporation experiment mirrors previous experiments and validations (see Taverna et al., 2012 and Wong et al., 2014) and allows us to conclude that automated injection does not perturb the microinjected cells (and the neighboring cells) more than manual injection. Based on the EdU incorporation, the microinjected cells are indistinguishable from non-injected cells. Furthermore, the addition of mRNA to the microinjection solution does not change EdU incorporation profile.

Despite the potential for high throughput with the robotic platform, "n" numbers remain very low for most experiments. It is also not clear how many slices were used in each experiment nor the reproducibility of the results with the use of more biological replicates.

We realized that the sentence was poorly phrased. The n value here refers to the number of neurons stained for Reelin, not the **total number of neurons** injected with the autoinjector. We stained only a subset of the slices for Reelin only a subset of neurons which is the number indicated in the original manuscript. To clarify, we have now added total n number of neurons injected, the n number of Reelin-stained neurons and the n number of Satb2-stained neurons in the figure legends. Further, we like to note that we intentionally separated the attempted microinjections by 30-40 μm to account for the complex shape as well as size of the neurons. This allowed us to trace the morphology of single isolated microinjection neurons. As a consequence of higher spacing, the total number of cells is bound to be on the low side.

Despite the seemingly low n number, we do believe the ability to microinjection neurons is a significant step as this was not possible with manual microinjection work we had previously attempted.

For instance, for mRNA injection (4H), 17 cells and for the quantification of reelin+ cells (6D), 12 cells were considered.

We note that we have previously used mRNA(s) microinjection for functional manipulation in a number of our previous studies (see: Taverna et al, 2012, Nat Neurosci; Wong et al, 2014, Nat Prot; Florio et al, 2015, Science; Kalebic et al., 2016, EMBO Reports; Tavano et al, 2018, Neuron). Thus, we only performed confirmatory experiments to demonstrate feasibility of mRNA microinjection using the Autoinjector.

Since we realized that these points might not be obvious for the reader, we decided to add an appendix in the user's manual where we explain which spacing to use for what type of injection and analysis, and how this impacts the expected number of injected cells (see "How one chooses the correct spacing between injections" in Appendix). We hope the reviewer, the readers and future users will find this helpful.

Further evidence is needed to support the claim that cell coupling is linked to the presence of apical attachment of the cells. As currently presented, it rather looks like coupling is highly correlated with targeting cells during the mitotic phase (which occurs apically) and could therefore reflect targeting two daughter cells in the process of separating. In fact, since apical progenitors in the VZ constitute a pseudostratified epithelium, it should be possible to target them when their cell bodies are away from the VZ (not in M-phase) and still visualize their connections.

We thank the reviewer for raising this point, that is mainly pertaining the analysis and interpretation of the two-cell-cluster data.

The reviewer raised two possible scenarios: targeting cells in M-phase proper, or targeting daughter cells in the process of separating, connected via the midbody bridge.

We would like to separately discuss these two very relevant points.

Targeting M-phase cells: targeting cells in M-phase using microinjection is in principle possible. However, our experimental data based on all the microinjection experiments we run so far suggest we never targeted cells in M-phase (Taverna et al, 2012 and Taverna, unpublished observations). This was observed for both manual and automated microinjection and it is consistent with the practical observation that cells in M-phase (even the one in 2-D culture) are difficult to target. We hypothesize that the spherical shape of the M-phase cell and the tension exerted by the mitotic spindle make the injection difficult and therefore makes cells in M-phase challenging to inject.

Targeting daughter cells in the process of separating, connected via the midbody bridge. The reviewer is raising an excellent point here. It is in principle possible that we target daughter cells that are still connected via the midbody bridge, with the nucleus far away from the apical surface.

This would result in two cell-clusters that are not coupled by gap-junctions, but simply via the midbody bridge.

This issue is addressed in **Appendix Figure S6** where we use a two-microinjection dyes paradigm. For all the coupling experiments, we injected a low molecular weight dye (A488) along with a high-molecular weight dye (DxA555).

The rationale is that daughter cells (connected via the midbody bridge) would be positive for both dyes. In case the two cells are coupled via gap-junctions, then only one (the targeted cell) should be positive for the high-molecular weight dye (see scheme in **Appendix Figure S6**).

In the analysis we showed, we restrict our analysis only and exclusively to the two-cells cluster where only one cell is positive for the high-molecular weight dye (DxA555), so that we automatically exclude those cases where the two cells are daughter cells (likely connected with the midbody bridge). In our hands, the vast majority of two cells cluster falls into this category. In all the experiments we run, we have found only 2 clusters that were positive for both low and high-molecular weight dyes.

The claim of high versatility of the technique should be backed up by data showing its usefulness for targeting adult brain and different types of tissue (other than neural), and at least one other species (zebrafish, chick, drosophila or c.elegans as suggested by the authors). Same applies to the versatility of substances that can be delivered. Can proteins or drugs be also microinjected successfully with the technique?

Targeting adult brain and different types of tissue, and other species. We think the points raised by the reviewer represents the future avenues of research and application for our system. However, we think that showing the applicability of the autoinjector in such a wide list of tissues/species is beyond the scope of our current work and of our paper.

Versatility of substances. We fully agreed with the Reviewer and decide to address the versatility of substances in the **Extended View Figure 1** in which we show the efficiency of injection for the following classes of chemicals

- (i) injection dye and their combination
- (ii) recombinant proteins
- (iii) drugs
- (iv) mRNA

The quantifications show that the efficiency varies depending on the chemical one injects. Specifically, the efficiency is mainly depending on the intrinsic tendency of the chemical to aggregate. The tendency to aggregate impacts directly the likelihood of the pipette of getting clogged, and indirectly the efficiency of microinjection. These results are in complete agreement with what is known from microinjection experiments in cell lines in 2D culture.

Minor comments:

-It is not currently clear in the text how much time does the optimization of the parameters take, especially for an untrained user. In this same line, does the optimal pressure depend on the type of cell or type of tissue to be microinjected? How thick should the injection capillaries be and does this, again, depend on cell type/tissue? A list of general guidelines or a table with the parameters used for different cells/substances, would be very useful.

As with any technique, there will be some optimization that will be required. While we have strived to demonstrate the versatility and applicability of the Autoinjector in a variety of contexts, it is not possible to anticipate every potential application and demonstrate it in one study. Second, while we have automated the process of microinjection, factors such as geometry of pipette, tissue health, experimental conditions, reagents being injected all affect microinjection yield and efficacy. These will need to be optimized by the individual laboratories. We have previously published an exhaustive protocols paper (**Wong et al Nature Protocols 2014**) that discusses these parameters and ways to troubleshoot them. Further, in this manuscript we have now included an appendix that provides the readers with practical tips for starting Autoinjection. This Appendix provides tips on obtaining an ideal pipette for microinjection, issues encountered when microinjecting viscous reagents and some considerations for determining the spacing between microinjections. The settings and conditions used in this study can be used as a good starting point when attempting to microinject single cells and can later be modified as desired and required by the end users.

A high magnification image of a single cell being injected would really help having a sense of the

invasiveness of the technique.

We unfortunately cannot provide what the reviewer is requesting for two technical reasons.

(i) We currently use a phase contrast 2x objective to image a 250-300 μ m thick brain slice.

The resolution of the objective does not allow to clearly visualize a single cell in the tissue.

(ii) The microinjection pipette moves at \sim 100 μ m/sec. Given the thickness of tissue, accurate visualization of single cell penetration will take a completely different optical imaging setup.

When assessing the coupling between cells (page 12), it is stated that the tissue is fixed immediately after microinjection. The timing should be given in a more specific manner as the average time from injection to fixation (the first cells injected had longer periods until fixation).

The tissue is typically fixed \sim 15 minutes after microinjection. This information has now been added in the material and methods.

The authors show nicely how the lineage progression of the microinjected cells in the slices can be followed over time. How does this compare to the timing of lineage progression of progenitors in the intact brain?

The timing of linear progression of progenitors is similar to the cell cycle duration measured and reported by Calegari and Arai in the developing dorsal telencephalon of the mouse at E14.5 (see Arai et al., Nature Communications, 2011; Calegari et al., J. Neurosci., 2005)

Can the robotic platform aid the injection of multiple substances in the same slice? And do the authors envision a possibility of eventually targeting single cells in living organisms (not necessarily mice)?

We think it is possible to inject different substances in the same slice. We also think it would be possible to inject tissue other than the brain. In general, we think Autoinjector can be used to target any tissue, provided the surface is exposed to the pipette. We already obtained promising results by targeting the embryonic mouse skin (Elena Taverna, unpublished observation).

The text in the abstract needs to be revised, as well.

We have now edited the abstract to contain shorter sentences to enhance readability.

Reviewer #3

Reviewer's Comment:

The manuscript by Shull et al described an automated microinjection platform, called Autoinjector, that consists in using images acquired by a microscope to guide a microinjection pipette. This customized platform allows significant gain of time and precision, with an increased number of cells successfully injected within a shorter period. In addition, while manual injection needs a lot of practice to be mastered, Autoinjector can be operated by an inexperienced user. Although this platform brings many advantages and although authors provide a user manual, implementation of the technology in other laboratories may be limited to qualified users (expertise in microscopy as it requires modification of the standard set-up). In addition, to better assess the performance of the Autoinjector, authors should address the following points

Authors have tested several depths for microinjection in APs (ranging from 10 to 35 μ m). In Figure 3A, Authors show a 35% success rate with the following parameters: depth 10 μ m; pressure: 75mbar. In Figure 3B and 3C, they show a rate of 68% and 43%, respectively, using the exact same conditions. Could the authors comment on these discrepancies? Have the authors really assess the variability of the platform (from one day to the other, etc...)?

The depth-pressure conditions in Figure 3A and 3B are not the same. In Figure 3A the depth was ranging from 10 to 20 μ m (we did not know yet the optimal depth, that's why we used a range).

The experiments in 3A and 3B were performed in two different days, as the results in 3A were necessary to inform the choice of the parameters in 3B.

The experiments in 3C were run to assess specifically the differences between manual and automated injection, the average is made by using several of experiments, performed in different days where we injected different chemicals. Of note, the data on efficiency in 3A and 3B were not

added to the graph in 3C. The lower efficiency in 3C precisely reflects the variability in the efficiency of the platform, as pointed out by the reviewer.

By injecting the APs, one would expect labelling of the whole cell including the glial process that extend from the ventricle to the pia. Authors do show such a staining but only in one cell (fig3B). Why processes are not labelled in Figure 3A and B?

There are two main reasons for that, one is biological, the other technical.

Biological reason: Apical progenitors in the VZ include (i) apical radial glia, featuring a long basal process that reaches the basal lamina and (ii) short neural precursors, lacking the basal process. Given the very good spatial resolution we can achieve with the autoinjector, we can visualize both of these populations (see also **EV Figure 2**, panel A, for a direct comparison of a aRG and SNP side by side).

Technical reason: the 300µm-thin slices are fixed and then are re-sliced into 50µm slices in order to perform immunofluorescence analysis. Although we try our best to properly orient the 300µm-thick

slice before cutting it, chances are very high that the re-slicing will cut the aRG in two, eliminating the basal process. In agreement with that, we often find slices where we visualize the basal process only, and the correspondent cell bodies is on a different slice. All these cases can be easily recognized as the what is left of the basal process abruptly ends at the boundary of the 50µm slice.

Reviewer's Comment:

One advantage of microinjection is injection of non-charged molecule. It would have been really valuable to show that Autoinjector also increases the rate of success of injection with such molecule.

We note that the microinjection dyes are almost all non-charged molecules. To address the Reviewer comment further, we have quantified the efficiency of microinjection for different chemicals. The results are shown in **EV Figure 1**. We tested the following classes of chemicals:

- (i) injection dye(s) and their combination
- (ii) recombinant proteins
- (iii) drugs
- (iv) mRNA

Authors claim that injection of a gap junction permeable dye might help defining how many cells are coupled via gap junction. Could the volume of injected dye influence the results? It is quite puzzling to see similar fluorescence intensity in cells in small and large clusters (fig5a versus 5C). If the same volume has been injected, we would expect a decrease of the signal in the large cluster. The signal of the gap-junctional permeable dye is the result indirect immunofluorescence using an antibody anti-A488. The enhancement with the primary and secondary might mask the differences between cells. Also, in the images we show the laser power was set to have comparable signals in all the clusters.

Quid of cell death over time? Does use of Autoinjector affect survival in the same way than manual injection?

Following the reviewer's comment, we quantified the number of picnotic nuclei in

- in vivo
- non injected slices (24h culture)
- non-injected cells in the microinjected-area in a slice injected with dye (24h culture; manual injection)
- non-injected cells in the microinjected-area in a slice injected with dye (24h culture; automated injection)
- non-injected cells in the microinjected-area in a slice injected with dye and mRNA for RFP (24h culture; automated injection)

The results are shown in **Appendix Figure S4**.

As for the cell death in microinjected cells, this parameter cannot be assessed in a faithful way in microinjected cells after 24h in culture. It is known from work with microinjection of cells in petridish that cells damaged by microinjection are usually dying in the first 2-3hours after microinjection. If cells manage to overcome this "critical phase" then their behavior and survival is comparable to that of non-manipulated cells.

We therefore decided to use EdU incorporation to better assess the impact of microinjection on cell viability, for both microinjected and non-microinjected cells. We performed and quantified EdU incorporation for 24h in:

- non injected slices
- microinjected cells in a microinjected slice (manual injection)
- microinjected cells in a microinjected slice (automated injection)
- non-injected cells in the microinjected-area in an injected slice (manual injection)
- non-injected cells in the microinjected-area in an injected slice (automated injection)

The results are shown in **Appendix Figure S5**.

The EdU incorporation experiment mirrors previous experiments and validations (see Taverna et al., 2012 and Wong et al., 2014) and allows us to conclude that automated injection does not perturb the microinjected cells (and the neighboring cells) more than manual injection. Based on the EdU incorporation, the microinjected cells are indistinguishable from non-injected cells. Furthermore, the addition of mRNA to the microinjection solution does not change the EdU incorporation profile.

Authors are used 200 to 500 μ M-thick organotypic section. Is the thickness of the section influencing the yield of injection?

The thickness of the slice is not a factor that affects yield of injection, but does affect slice survival. In our hands, the slice survival is better for 300-500 μ m slices compared to thinner slices (>150 μ m). Further, thicker slices offer bigger area of apical surface to inject. Thus, in mice, we use 300 μ m thick slices, while for the human tissue we use a range from 300 to 500 μ m (the slicing in this case is done by hand, preventing a precise control over the thickness of the slice). We added a discussion on slice in the appendix.

Fig4C: not clear if Daughter cells 3 and 4 are negative for both Tbr2 and Tbr1. If so, what is the fate of those cells, APs? Their localization argues against this hypothesis. Could the authors comment? In fig4, it would have been interesting to provide a sax2 or Pax6 staining. On the same line, in Figure6C, an upper layer (cuX1) staining is missing.

Authors' Response:

Daughter cells 3 and 4 are mild positive for TuJ1 and Tbr2 (the signal might not be optimal in the printed version). Based on their pattern of staining, morphology and position, we concluded that these cells are newborn neurons.

Sox2: We have performed new experiments to quantify Sox2 positivity of microinjected cells and their progeny at 0h, 24h and 48h. These new data along with the quantification, are included in **EV Figure 2** in the revised manuscript.

As the yield of injection is pretty low in hindbrain and human APs, what is the added value of Autoinjector?

We included this data to demonstrate versatility and generalizability of the Automated microinjector. We show only one cell for hindbrain and human to that we can achieve a really good single cell resolution. The pictures are meant to show proof of principle and one cannot extrapolate efficiency from these pictures.

For the hindbrain, panel A is an image taken at the microinjection scope and show a quite high number of cells, ruling against a low efficiency.

As for the human tissue, it is possible that the efficiency is low. Unfortunately, the rarity and availability of human fetal tissue precluded us from optimizing any parameter.

When performing injection from the basal surface, do the author target APs (through basal feet of the radial process)? At which rate?

In all the experiments we performed in the basal surface, we saw only one positive AP. At this point, this is only an anecdotal observation. Given the very low chance of targeting an AP using basal injection, we feel it is unfair to report it as a possibility.

As for the reason, it might be that the basal process is too thin and possibly not enough stable to be targeted using the microinjection pipette.

Minor points:

- First sentence of the abstract may be revised.

This has been revised.

- reference to Figure 5G and I are missing.

Thank you for pointing this out. These are now referenced in the revised manuscript.

- page 12: should be 5J instead of 5K

This has been updated in the main manuscript.

- page 13: should be 5K instead of 5J

This has been updated in the main manuscript.

- discussion page 15: why authors are mentioning injection across 1mm while they performed injection with depth ranging from 10 to 25um (for APs)?

The injection was performed at a depth of 10 to 25um across a large portion of the apical surface (1 mm). With this sentence we wanted to describe how the Autoinjector could perform fine injection depth control (micron scale) without compromising area coverage (on the scale of mm).

2nd Editorial Decision

8 July 2019

Thank you for the submission of your revised manuscript. We have now received the enclosed reports from the referees that were asked to assess it. 2 referees still have a few minor suggestions that I would like you to incorporate before we can proceed with the official acceptance of your manuscript.

A few more changes will also be required:

- Please confirm that you want to keep the synopsis image with the current text. I think it is unclear what the text means.

- Please send us all movie files as zipped files of the movie together with their legends. The movie files need to be called Movie EV1, 2, 3, etc.

- The callouts for figures 3F, 4A-C, 5H, I and for all EV figure panels are missing. Fig 2A is called out after Fig 3. Please correct and add all callouts.

- The inset images of Fig 6C and 6D do not seem to be the same cells as in the overview image. 6C might be flipped and 6D looks like a different cell. Please clarify.

- Please send us up to 5 keywords.

- I attach a word file to this email with requested changes to your manuscript text file. Please make these corrections in the final manuscript word file. You can also accept the changes and upload the attached file.

REFeree REPORTS

Referee #1:

Shull and colleagues addressed all major points that were raised in the initial review of this paper well. They provide compelling responses and revised the manuscript as necessary. Altogether the manuscript provides a valuable resource for the community.

Referee #2:

The authors address most of the points raised by this reviewer, including the most crucial ones on cell viability, versatility of the technique and the use of slightly misleading language regarding the use of the technique in intact tissue. I therefore support the publication of the manuscript. Please revise some small errors and typos, e.g.: Grammatically incorrect sentences such as in the abstract: The Autoinjector successfully targetS... ; in page 11: The majority OF the microinjected cells... Also an image seems to be missing in supplementary figure 2, panel B.

Referee #3:

Substantial modification have been made in this revised version of Shull et al manuscript. The authors have adressed most of the points that I raised. Although authors provide clear scientific arguments in the response to reviewers, I regret that authors make the choice to only cite the figure related to those key experiments (cell death, versatility using other chemicals or molecules) without giving any explanation. Some comments in the text would have been appreciated. however, this choice might have been driven by space constraints. I would suggest the authors to merge the two first paragraphs on page 10 (that pretty much say the same thing) to free space for additional comment on cell viability in the next paragraph. finally, concerning the EV Figure 2 added in the revised version, and the fact that they target both RGs and SNP with the automated injection : I am really surprise that they have such labelling of SNPs whose number is quite low in the mouse cortex. A containing with a SNP specific marker would help raising such conclusion. In this EV figure 2, the Dx-A555 MIP image in the panel B does not correctly appear. In conclusion, I would recommend this revised version to be published in EMBO Reports.

2nd Revision - authors' response

23 July 2019

Referee #1:

Shull and colleagues addressed all major points that were raised in the initial review of this paper well. They provide compelling responses and revised the manuscript as necessary. Altogether the manuscript provides a valuable resource for the community.
We thank Reviewer#1 for these comments.

Referee #2:

The authors address most of the points raised by this reviewer, including the most crucial ones on cell viability, versatility of the technique and the use of slightly misleading language regarding the use of the technique in intact tissue. I therefore support the publication of the manuscript. Please revise some small errors and typos, e.g.: Grammatically incorrect sentences such as in the abstract:

The Autoinjector successfully targetS... ;
We have now fixed the typo.

in page 11: The majority OF the microinjected cells...
We have now fixed the typo.

Also an image seems to be missing in supplementary figure 2, panel B.
The panel B in Supplementary Figure 2 was meant to contain only schematic. To avoid any confusion, we change the first sentence of Figure legend, that now reads: "The schematic shows the parameters defined by the user to create the final trajectory of the Autoinjector. "

Referee #3:

Substantial modification have been made in this revised version of Shull et al manuscript. The

authors have adressed most of the points that I raised. Although authors provide clear scientific arguments in the response to reviewers, I regret that authors make the choice to only cite the figure related to those key experiments (cell death, versatility using other chemicals or molecules) without giving any explanation. Some comments in the text would have been appreciated. however, this choice might have been driven by space constraints.

The experiments regarding cell death, versatility using other chemicals and their meaning for the users are discussed in the supplementary material and in the User Manual.

In addition, a paragraph about cell viability was added on page 10 (see below).

I would suggest the authors to merge the two first paragraphs on page 10 (that pretty much say the same thing) to free space for additional comment on cell viability in the next paragraph.

Following the Reviewer #3's suggestion, we have inserted a paragraph on page 10 where we elaborate more extensively on tissue and cell viability.

Finally, concerning the EV Figure 2 added in the revised version, and the fact that they target both RGs and SNP with the automated injection: I am really surprise that they have such labelling of SNPs whose number is quite low in the mouse cortex. A containing with a SNP specific marker would help raising such conclusion.

In our hands, the vast majority of cell that do not show a basal process are the result of a technical artifact: the cells have been cut during the slicing procedure. Only a minor fraction is represented by SNPs.

We apologize if our phrasing suggested otherwise.

In this EV figure 2, the Dx-A555 MIP image in the panel B does not correctly appear.

Some data must have gone missing during the processing and/or uploading.

We now re-uploaded EV Figure 2.

In conclusion, I would recommend this revised version to be published in EMBO Reports.

We greatly appreciated the conclusion of Reviewer #3.

Corresponding Author Name: Elena Taverna

Manuscript Number: EMBOR-2019-47880V1